# Functional differences between *TSHR* alleles associate with variation in spawning season in Atlantic herring

Junfeng Chen [1,8✉], Huijuan Bi[1], Mats E. Pettersson [1], Daiki X. Sato [1,2], Angela P. Fuentes-Pardo [1], Chunheng Mo[1,3], Shady Younis [1,9], Ola Wallerman[1], Patric Jern [1], Gregorio Molés [4], Ana Gómez[4], Gunnar Kleinau[5], Patrick Scheerer [5] & Leif Andersson [1,6,7✉]

The underlying molecular mechanisms that determine long day versus short day breeders remain unknown in any organism. Atlantic herring provides a unique opportunity to examine the molecular mechanisms involved in reproduction timing, because both spring and autumn spawners exist within the same species. Although our previous whole genome comparisons revealed a strong association of *TSHR* alleles with spawning seasons, the functional consequences of these variants remain unknown. Here we examined the functional significance of six candidate *TSHR* mutations strongly associated with herring reproductive seasonality. We show that the L471M missense mutation in the spring-allele causes enhanced cAMP signaling. The best candidate non-coding mutation is a 5.2 kb retrotransposon insertion upstream of the *TSHR* transcription start site, near an open chromatin region, which is likely to affect *TSHR* expression. The insertion occurred prior to the split between Pacific and Atlantic herring and was lost in the autumn-allele. Our study shows that strongly associated coding and non-coding variants at the *TSHR* locus may both contribute to the regulation of seasonal reproduction in herring.

[1] Department of Medical Biochemistry and Microbiology, Uppsala University, Uppsala, Sweden. [2] Graduate School of Life Sciences, Tohoku University, Sendai, Japan. [3] Key Laboratory of Birth Defects and Related Diseases of Women and Children, Sichuan University, Chengdu, China. [4] Department of Fish Physiology and Biotechnology, Instituto de Acuicultura Torre de la Sal, Consejo Superior de Investigaciones Científicas (CSIC), Castellón, Spain. [5] Charité – Universitätsmedizin Berlin, corporate member of Freie Universität Berlin and Humboldt-Universität zu Berlin, Institute of Medical Physics and Biophysics (CC2), Group Protein X-ray Crystallography and Signal Transduction, Berlin, Germany. [6] Department of Animal Breeding and Genetics, Swedish University of Agricultural Sciences, Uppsala, Sweden. [7] Department of Veterinary Integrative Biosciences, Texas A&M University, College Station, TX, USA. [8] Present address: Institute of Transformative Bio-Molecules (ITbM); Laboratory of Animal Integrative Physiology, Graduate School of Bioagricultural Sciences, Nagoya University, Nagoya, Japan. [9] Present address: Division of Immunology and Rheumatology, Stanford University, Stanford, CA, USA. ✉email: junfeng.chen@itbm.nagoya-u.ac.jp; leif.andersson@imbim.uu.se

Animals living at temperate latitudes rely on the photo-period (day length) to adapt their behaviors, such as reproduction[1], migration[2], molting[1], and hibernation[3], to seasonal changes. Breeding at a specific time of the year, known as seasonal reproduction, ensures that offspring are born when the environment is best suited for survival (e.g. food availability and moderate climate). Organisms that reproduce during spring with increasing photoperiod, such as birds[4] and some fish species[5,6], are denoted long day (LD) breeders, and those that breed during the decreasing photoperiod in autumn, like sheep[7], are short day (SD) breeders. Although the adaptive strategy of seasonal breeding is well known, the molecular mechanisms involved in photoperiodism have started to be uncovered only recently[8–10] and the underlying factors that distinguish LD and SD breeders remain unknown in any organism.

The mechanism underlying photoperiodism has been explored in mammals, in which light is sensed by the retina and the signal transmitted to the pineal gland via the suprachiasmatic nucleus (SCN) where the circadian pacemaker is located[10,11]. The photoperiodic information is then reflected by the secretion profile of melatonin from the pineal gland, where melatonin exhibits a longer secretion peak during short days and a shorter peak when the day length is long. Melatonin subsequently regulates the secretion of thyrotropin (TSH) in pars tuberalis of the pituitary gland, which further acts on the TSH receptor (TSHR) on the surface of ependymal cells in the hypothalamus to regulate *DIO2* expression through a TSHR-Gsa-cAMP signaling pathway[11]. *DIO2* encodes type 2 deiodinase, which converts the prohormone thyroxine (T4) to bioactive triiodothyronine (T3). Comparison of the photoperiodic signaling mechanisms among mammals, birds, and fishes indicates that the downstream regulation of TSH on seasonal reproduction appears to be widely conserved in vertebrates[10]. However, the light input pathways are diverse across taxa. In fish, the entire photoperiodic signaling pathway from light entry to neuroendocrine output is proposed to be integrated into saccus vasculosus (SV), a fish-specific organ in the caudal hypothalamus[12]. However, the exact molecular mechanisms how the photoperiodic information is translated to reproductive output are still poorly understood.

Atlantic herring (*Clupea harengus*) is one of the most abundant fish species on Earth[13] and exhibits seasonal reproduction predominantly in spring and autumn. This species provides a unique opportunity to dissect the mechanisms controlling timing of reproduction due to the variation within species rather than between species which allow powerful genetic analysis. Our previous whole-genome comparisons have demonstrated that there is minute genetic differentiation between spring- and autumn-spawning herring at selectively neutral loci but striking genetic differentiation at about 30 loci[14–16]. The minute genetic differentiation at neutral loci is explained by a lack of complete reproductive isolation between spring and autumn spawners and minute genetic drift due to the huge population sizes. The most differentiated region between spring and autumn-spawning Atlantic herring overlaps *TSHR*[15,17,18] (Fig. 1a). Two non-coding SNPs, one upstream of the promoter region and the other in intron 1, and two missense mutations, Q370H and L471M, stood out as the most strongly associated with the phenotype.

The aim of the present study was to further characterize the *TSHR* locus in herring by genetic and functional analysis. We describe, two additional sequence variants distinguishing the spring and autumn haplotypes, (i) a loss of a retrotransposon sequence upstream of *TSHR* and (ii) a copy number polymorphism at the C terminal end of the *TSHR* transcript. We show using mutagenesis and transfection experiments that the L471M substitution in the spring-allele results in enhanced constitutive cAMP signaling, and the 5.2 kb retrotransposon variant upstream of *TSHR* near an open chromatin region may affect *TSHR* expression.

## Results

**Strong genetic differentiation at the *TSHR* locus between spring- and autumn-spawning herring.** A genome scan comparing seven populations of spring- and autumn-spawning herring revealed that the most strongly differentiated region occurs in the vicinity of the *TSHR* locus on chromosome 15 (Fig. 1a). To further characterize the genetic diversity at this locus we calculated nucleotide diversity ($\pi$) and Tajima's D per population, and the delta allele frequencies (dAF) between the two groups (Fig. 1b, c). The dAF on chromosome 15 peaks precisely at the *TSHR* locus consistent with the assumption that *TSHR* is the target of selection (Fig. 1b). Furthermore, the data were consistent with the presence of selective sweeps at the *TSHR* locus in both autumn- and spring-spawning herring as the nucleotide diversities are significantly lower in the *TSHR* region than they are in the rest of the genome (Fig. 1c). The sweep is most obvious in spring-spawning herring which has lower nucleotide diversity and more negative Tajima's D in this region (Fig. 1b). Thus, spring-spawning herring displayed a more pronounced footprint of selection, indicating a more recent selective sweep, consistent with previous findings[15,17].

**Characterization of three highly differentiated *TSHR* coding variants, two missense mutations and a copy number polymorphism.** A multi-species amino acid sequence alignment of TSHRs showed that the two highly differentiated missense mutations in Atlantic herring, Q370H and L471M, correspond to Q359 in the typical extracellular hinge region and F461 in the second transmembrane helix (TMH2) of human TSHR[19] (Fig. 2a, Supplementary Fig. 1). Several other fish TSHRs possess a conserved proline (P) at the site corresponding to residue 370 in herring, glutamine (Q) is present in mammalian orthologs and in the autumn-spawning Atlantic herring allele while the spring-spawning allele has histidine (H) at this position. The L471M substitution is located at a position where all known mammalian TSHRs exhibit phenylalanine (F) in contrast to leucine (L) in most fish species including autumn-spawning herring, which has been replaced with methionine (M) in the spring-spawning *TSHR* allele.

We identified a third highly differentiated coding variant by analyzing the two *TSHR* alleles in the genomic DNA of a heterozygous individual used for building the reference assembly ASM96633v1[15]. An in-frame 66 bp fragment coding for 22 amino acids (22aa) located seven residues before the stop codon displayed a copy number variation, the amino acid sequence of the repeat is shown in Fig. 2a. Individual genotyping showed that 84% of the spring spawners were homozygous for three copies of this repeat ($n = 45$) while 91% of the autumn spawners were homozygous for the allele with five copies or heterozygous for alleles with five and four copies (n = 67). Alignment of the herring TSHR protein sequence to orthologs in other species revealed an extended C-terminus due to this unique 22aa repeat. The repeat contains five negatively charged glutamic acid residues (E) and four potential threonine phosphorylation sites (T) (Fig. 2a). C-terminal receptor phosphorylation is a prerequisite for the internalization of G protein-coupled receptors (GPCRs)[20]. Thus, copy number variation of this repeat containing putative threonine/serine phosphorylation sites might result in a difference in the internalization and desensitization rate between the two allelic variants.

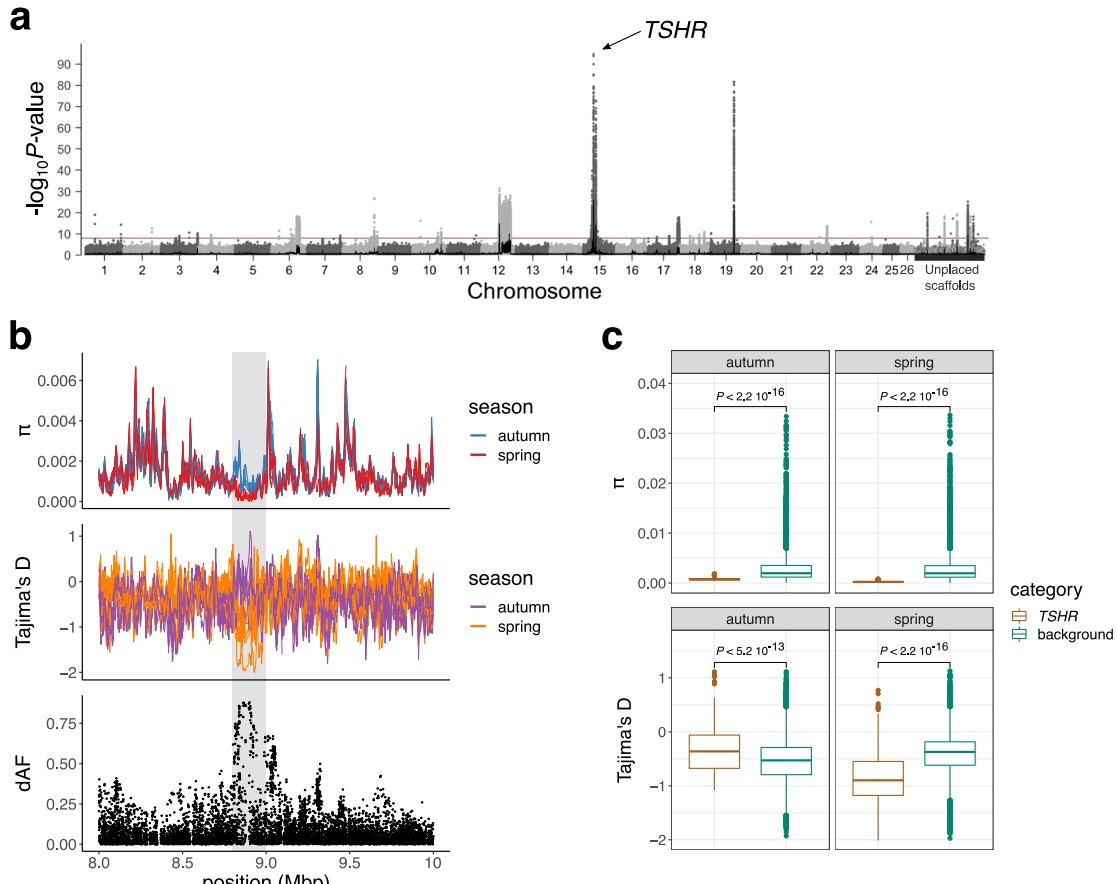

**Fig. 1 Genetic diversity across the *TSHR* region in spring- and autumn-spawning Atlantic herring. a** Genome scan based on 6.06 million SNPs for the identification of divergent genomic regions between seven spring- and seven autumn-spawning populations (based on data from Han et al.[16]). The y-axis represents the significance value (-log₁₀P-value) per SNP in a $X^2$ test comparing the allele frequencies in spring- and autumn-spawning populations. Each dot corresponds to a single SNP, and different shades of gray were used to distinguish SNPs in consecutive chromosomes. The horizontal red line indicates the significance threshold based on Bonferroni correction. **b** Genome-wide diversity statistics π and Tajima's D, and absolute allele frequency differences (dAF) of seven spring- and seven autumn-spawning herring pools in chr 15: 8–10 Mbp. The location of *TSHR* is denoted with a shadowed gray area (chr 15: 8.85–8.95 Mbp). Each line corresponds to a single pool, and the two colors distinguish autumn- and spring-spawning populations. **c** Comparison of the π and Tajima's D distributions between SNPs on chromosome 15 located outside (214 635 SNPs) and inside (8.85–8.95 Mbp, 215 SNPs) the *TSHR* region. The P-values were obtained from the Wilcoxon test applied to estimate the statistical significance of the mean differences between SNPs inside and outside the *TSHR* region for the spring- and autumn-spawning populations.

**Characterization of a 5.2 kb structural variant upstream of *TSHR*.** A comparison of the genomic region harboring the *TSHR* locus between the two reference assemblies of Atlantic herring (ASM96633v1[15] and Ch_v2.0.2[18]), which represent the autumn and spring haplotypes, respectively, revealed a 5.2 kb insertion in the spring allele about 2.15 kb upstream of the transcription start site (TSS) of *TSHR* (Fig. 2b). Dot plot analysis of the 10 kb region spanning the insertion identified two 470 bp LTR-like fragments flanking the structural variant, suggesting the presence of an LTR retrotransposon at this locus (Supplementary Fig. 2a). Repeat masking by CENSOR[21] annotated the retrotransposon as a BEL/Pao element and its surrounding sequences as a non-LTR retrotransposon Rex1 (Fig. 2b). Blast analysis identified eight additional BEL/Pao copies with >98% nucleotide identities in the herring genome. A 3543-bp ORF coding for 1,180 aa was predicted for the BEL/Pao element, which was best aligned to residues 420–950 and 1270–1950 of a zebrafish uncharacterized protein (NCBI ID: XP_021336828) with retrotransposon features. Based on the predicted domains of this zebrafish protein, the herring BEL/Pao element contains part of the Pao retrotransposon peptidase domain and the intact integrase core domain, but lack the entire reverse transcriptase.

Genotyping of this 5.2 kb structural variant exhibited an extremely strong association with spawning seasons in herring; spring spawners were nearly fixed for an allele harboring the BEL/Pao and Rex1 elements, whereas the BEL/Pao element has recombined in the autumn allele leaving a solo LTR and Rex1 (Fig. 2b). Sequence analysis among 10 spring spawners revealed four mismatches between the two LTRs (Supplementary Fig. 2b). This sequence divergence of about 1% (4/470 bp) indicates that the insertion of BEL/Pao into Rex1 occurred millions of years ago. This was confirmed by long-range PCR showing the presence of the BEL/Pao insertion also in the Pacific herring (*Clupea pallasii*) that diverged from Atlantic herring about two million years before present[15]. There is a single mismatch between the solo LTR in the autumn allele and the right LTR in the spring allele whereas there are five mismatches to the left LTR, implying that the right LTR contributed most to the solo LTR that was left behind when the BEL/Pao element recombined in the autumn allele. The presence of only a single mismatch between the solo LTR in the autumn allele and the right LTR in the spring allele indicates a recent origin of the allele lacking BEL/Pao compared with the original insertion of the BEL/Pao element upstream of *TSHR*. Furthermore, sequence analysis of BEL/Pao in Pacific

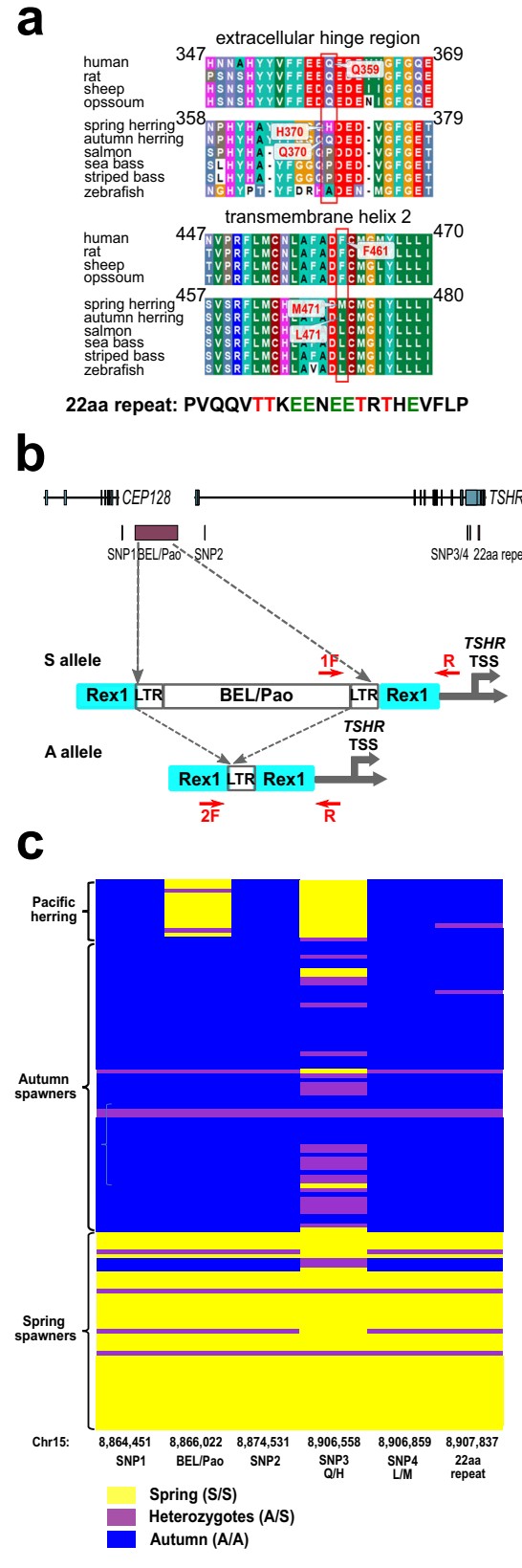

**Fig. 2 Genetic variants at the *TSHR* locus strongly associated with seasonal reproduction in Atlantic herring. a** Interspecies sequence alignment of the TSHR domains including the two major missense mutations in herring *TSHR*. Residues with hydrophobic (green), positively charged (blue), or negatively charged (red) side chains are highlighted. The amino acid composition of the 22aa repeat at the C terminus of herring TSHR is also shown with the potential phosphorylation threonine (T) sites highlighted in red and negatively charged glutamic acid (E) highlighted in green. **b**) Comparison of the genomic region harboring the *TSHR* locus between the Spring and Autumn haplotypes reveals a 5.2 kb insertion in the Spring allele about 2.15 kb upstream of the *TSHR* TSS. The Spring haplotype harbors both Rex1 and BEL/Pao elements while the BEL/Pao internal sequence with its left LTR (indicated between the two arrowed dash lines) is absent in the autumn haplotype. Locations of genotyping primers for this structural variant are indicated by red arrows. **c** Haplotype analysis of six differentiated sequence variants among 45 spring-spawning, 67 autumn-spawning Atlantic herring and 13 Pacific herring. The genomic location of each sequence variant is indicated.

**Haplotype analysis**. To determine the strength of association between candidate causal variants and phenotype, the six *TSHR* genetic variants, spanning a 35 kb region, showing strong genetic differentiation between spring and autumn spawners were genotyped in 45 individuals from six different spring-spawning populations and 67 individuals from eight autumn-spawning populations from the Baltic Sea, North Sea, East and West-Atlantic Ocean. This analysis revealed strong linkage disequilibrium among the various genetic variants, with two major haplotypes, spring (S) and autumn (A), which were strongly associated with spring and autumn spawning, respectively. The majority of autumn spawners were homozygous A/A whereas spring spawners were homozygous S/S (Fig. 2c). The only exception to this rule was SNP3, the Q370H substitution, which was polymorphic among A haplotypes, suggesting that it is of more recent origin or that it is under balancing selection.

A comparison with the *TSHR* haplotypes in Pacific herring shows that the S allele in Atlantic herring is associated with the derived allele at 4 of the 6 strongly differentiated sequence polymorphisms (Fig. 2c). The exceptions are the Q370H missense mutation, consistent with a more recent origin on the A haplotype, and the loss of the BEL/Pao internal sequence. Two Pacific herring individuals were heterozygous for an allele lacking the BEL/Pao internal sequence. The sequence of this solo LTR was identical to the LTR sequences of the complete BEL/Pao and thus different from the solo LTR in the Atlantic autumn allele (Supplementary Fig. 2b). This suggests that the BEL/Pao internal sequence must have been lost independently in Pacific and Atlantic herring after the split from a common ancestor. We have no phenotype data as regards spawning time for the Pacific herring included in this study as they were caught outside the spawning season, but Pacific herring is primarily spring spawning.

**ATAC-seq identifies an open chromatin region close to the 5.2 kb structural variant**. ATAC-sequencing (ATAC-seq) was performed to explore the regulatory regions at the *TSHR* locus using two herring brain and brain area containing hypothalamus and the fish-specific saccus vasculosus (BSH) samples from spring-spawning Baltic herring collected during spawning. All four samples showed a single consistent ATAC-seq peak at the *TSHR* locus, which might harbor a critical element regulating the expression of *TSHR* in these two tissues (Fig. 3a). The identified peak (Chr15: 8,863,792–8,864,084 in brain and 8,863,798–8,864,102 in BSH) was located 350 bp upstream of the highly differentiated SNP1, 1.1 kb

herring showed that the two flanking LTRs are identical. The LTRs present in Pacific herring do not carry any unique SNPs and are therefore expected to represent the LTR sequence present in the common ancestor of Pacific and Atlantic herring (Supplementary Fig. 2b).

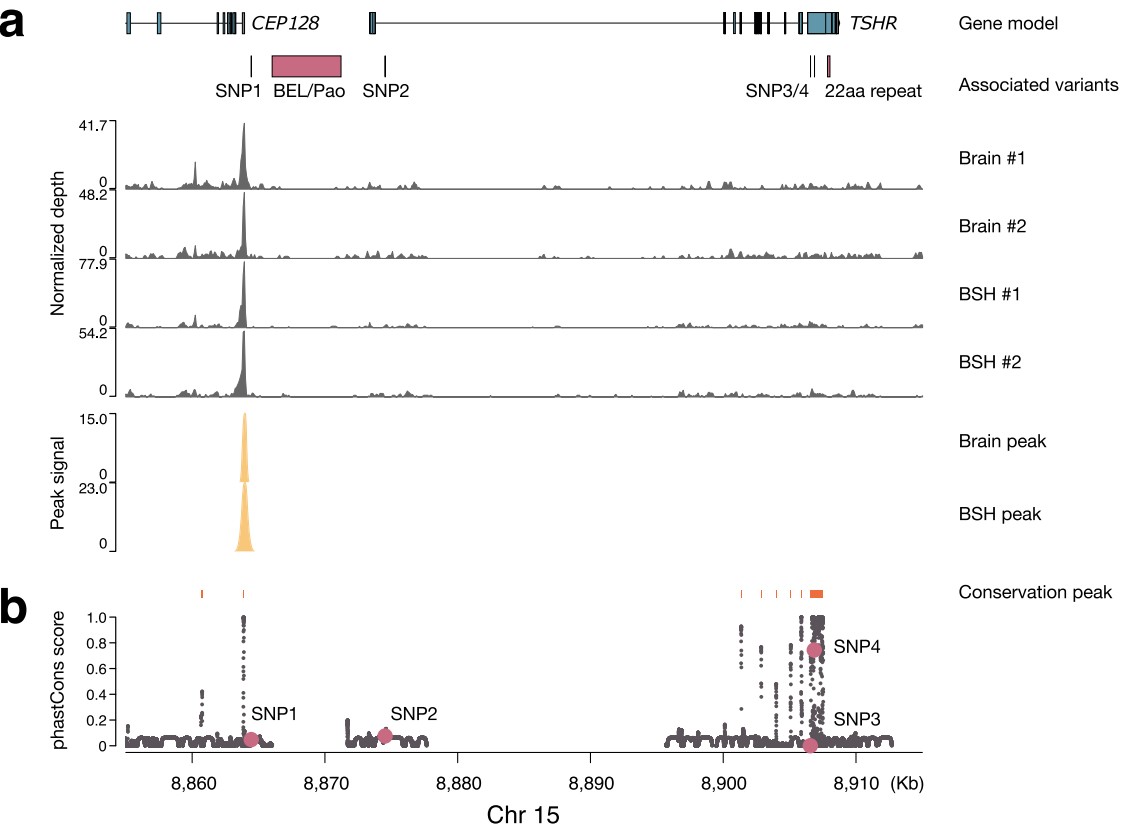

**Fig. 3 ATAC-seq and sequence conservation analysis for the herring *TSHR* locus. a** ATAC-seq signals in herring hypothalamus and saccus vasculosus (BSH) and brain without BSH (brain). Genomic locations of two annotated genes at this locus, *CEP128* and *TSHR*, are indicated together with the six highly differentiated variants between spring- and autumn-spawning herring, including two non-coding SNPs (SNP1 and 2), two coding SNPs (SNP3 and 4), the 5.2 kb BEL/Pao structural variant and the 22aa copy number variant at the C terminus of herring TSHR. **b** Sequence conservation represented by the phastCons score calculated using data from 12 fish species.

upstream of Rex1 and 1.9 kb upstream of the 5.2 kb BEL/Pao insertion. The distance from the ATAC-seq peak to the *TSHR* TSS was strikingly different for the spring and autumn haplotypes (9.3 and 4.1 kb, respectively) due to the presence/absence of the BEL/Pao retrotransposon, which might affect *TSHR* expression.

**Tissue expression patterns of *TSHR*, *TSHB*, and *DIO2*.** Hypothalamus is the regulatory hub for photoperiodism in mammals and birds, and BSH has this role in fishes[10]. Several key genes involved in the photoperiodic signaling pathway, including *TSHR*, *TSHB,* and *DIO2*, are highly expressed in this tissue[9,11,12]. Expressions of these three genes among various tissues were explored using qPCR in spring-spawning Atlantic herring. All three genes were expressed at the highest levels in brain or BSH, indicating a key role of these tissues in photoperiodic regulation in Atlantic herring (Fig. 4a–c). However, each gene had a tissue-specific expression profile. *TSHR* was more generally expressed among multiple tissues with the highest level in BSH and a moderate level in gonad, heart, and brain, suggesting its broad spectrum of biological functions[22] (Fig. 4a). Expressions of *TSHB* and *DIO2* were more restricted to the brain and BSH (Fig. 4b, c).

**Functional analysis of herring TSHR.** Gα$_s$-cAMP signaling activities of the spring (S) and autumn (A) TSHR variants were investigated with transfection experiments using the fish Epithelioma Papulosum Cyprini (EPC) cell line and with constructs representing the S and A *TSHR* variants, including two missense mutations and the 22aa copy number polymorphism (S: H370, M471 and three copies of the 22aa repeat; A: Q370, L471 and five

copies of the 22aa repeat). The level of cell surface expression of the two receptor variants was compared using flow cytometry before functional analysis. Only marginal differences in fluorescent intensity were observed between cells transfected with different alleles, indicating a similar level of cell surface expression (Supplementary Fig. 3).

pGL4-CRE-dual-luciferase assays were performed to compare the Gα$_s$-cAMP signaling activities, where cells expressing the S variant exhibited a 2.2-fold increase in constitutive cAMP signaling compared with the A variant (Fig. 5a). To compare the TSH-induced signaling of the two TSHR variants, we produced recombinant herring single-chain (sc) TSH in CHO cells. Western blot analysis of the concentrated culture media detected with antisera against the sea bass (*Dicentrarchus labrax*) alpha subunit revealed the presence of scTSH as a single band of 45–50 kDa (Fig. 5b). In peptide-N-glycosidase F (PNGase F) treated media, scTSH was detected as a band around 37 kDa, suggesting that the produced scTSH is N-glycosylated (Fig. 5b, lane 1). We treated EPC cells expressing S or A herring TSHR with different dilutions of the concentrated media for 4 h, cells transfected with empty pcDNA3.1 plasmid did not display clear response to the TSH induction. However, the intracellular cAMP levels were dramatically enhanced in cells transfected with the S and A variants in a dose-dependent manner, and the S variant consistently showed on average a 20% stronger cAMP signaling activity than the A variant (Fig. 5b).

The TSHR-Gα$_s$-cAMP signaling is involved in the regulation of *DIO2* expression in both mammals and birds[9,23]. 5′ RACE was performed to determine the TSS of herring *DIO2* and two cAMP response elements (CRE: TGACGTCA) were identified within the

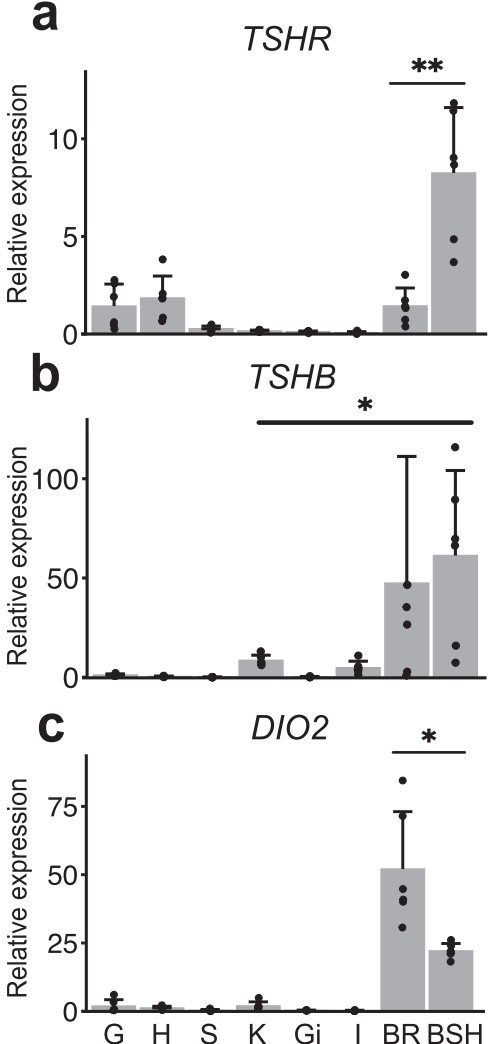

**Fig. 4 Tissue expression of three key genes (*TSHR*, *TSHB* and *DIO2*) involved in the photoperiodic signaling pathway examined by quantitative PCR. a** *TSHR*. **b** *TSHB*. **c** *DIO2*. Tissues that were used in the experiment include gonad (G), heart (H), spleen (S), kidney (K), gills (Gi), intestine (I), hypothalamus and saccus vasculosus (BSH), and brain without BSH (BR). The average expression level of each gene in the gonad is assumed to be 1, error bars represent the SDs calculated from six biological replicates ($N = 6$). Unpaired two-tailed Student's $t$ test was used for the statistical analysis. *: $P < 0.05$, **: $P < 0.01$.

720 bp promoter region, indicating that the TSH-TSHR-cAMP-DIO2 signaling pathway is conserved in Atlantic herring (Supplementary Fig. 4).

In conclusion, these studies showed that the spring TSHR allele is associated with both a higher constitutive activity (2.2-fold change) and a slightly increased maximum of TSH-induced cAMP signaling (1.2-fold enhancement) compared with the signaling properties of the autumn allele.

**The L471M substitution causes an enhanced constitutive TSHR signaling.** cAMP signaling activities were compared using eight herring TSHR constructs with different combinations of the three coding variants (the two missense mutations and the C-terminal repeat) to determine their individual effects on TSHR signaling (Fig. 5c). Cells expressing TSHR with the spring-version

M471 consistently displayed higher constitutive activity than those with the autumn-associated allele L471 irrespective of the other two coding variants (Fig. 5d), suggesting that the enhanced constitutive activity of spring TSHR was mainly attributed to the L471M substitution. However, we did not observe a clear association of any of the three variants with the differences in TSH-induced cAMP signaling (Fig. 5e). Thus, it is possible that the stronger TSH-induced activity of spring TSHR reflects its enhanced constitutive activity. Next, we mutated the two corresponding residues of human TSHR (residues: Q-F) to mimic the herring spring (H-M) or autumn (Q-L) alleles. Cells expressing human TSHR mimicking the spring allele consistently exhibited an increased constitutive activity (Fig. 5f). To further confirm the effect of spring M471, we only mutated the F461 of human TSHR to M461 and kept Q359 (Q-M combination), which again displayed an enhanced constitutive activity (Fig. 5f). These results demonstrated that L471M is associated with an enhanced constitutive activity both in the context of herring and human TSHR. Thus, this residue must be located in a functionally important domain in human and herring TSHR.

**Structural implications of the L471M substitution.** To investigate the potential underlying structural effects of the L471M substitution, we designed a structural TSHR homology model in complex with TSH and Gs-protein (Fig. 6). Due to the lack of suitable structural templates, variants in the hinge region (extracellular) and the copy number variation at the C terminus are not included in the model. First, the TSHR model indicated that the side chains residue at position 2.51 (M/L/F) in various TSHR orthologs are directed towards the membrane and do not participate in any ligand or Gs binding (Fig. 6a). Second, these side chains are tightly embedded between hydrophobic residues located in TMH1 and TMH2 (Fig. 6b) which presumably stabilizes the spatial orientation of both helices to each other. This hydrophobic patch deviates only at the central position 471 between the herring autumn and spring variants of TSHR (Fig. 6c). It is important to note that a methionine side chain is more flexible than the bulky aromatic phenylalanine side chain (in human TSHR) or the strong hydrophobic leucine side chains (in the autumn variant) (Fig. 6c). This higher side-chain flexibility and lower hydrophobicity of methionine instead of phenylalanine or leucine could lead to a less hydrophobic fixation in this region between TMH1 and TMH2.

**The L471M substitution and the ATAC-seq peak are both located in evolutionarily conserved regions.** To further explore the possible functional significance of sequence variant at the *TSHR* locus, the degree of sequence conservation was calculated using phastCons scores[24,25] on the basis of genomic sequences from 12 fish species. Multiple highly conserved regions were detected within the gene and both upstream and downstream of the coding sequence (Fig. 3b). Among the four differentiated SNPs, SNP4 resulting in the L471M substitution was located in a highly conserved 944 bp region with a phastCons score of 0.744, whereas the other three SNPs showed little conservation with phastCons scores of 0.048 for SNP1, 0.076 for SNP2 and 0 for SNP3. Furthermore, one of the conservation peaks (Chr15: 8,863,857–8,863,899) overlaps the ATAC-seq peak, supporting the functional significance of this region. These results further implicated that both the L471M substitution, which increases the constitutive activity using both human and Atlantic herring TSHR constructs, and BEL/Pao structural variant affecting the distance between a putative regulatory element and *TSHR* are functionally important.

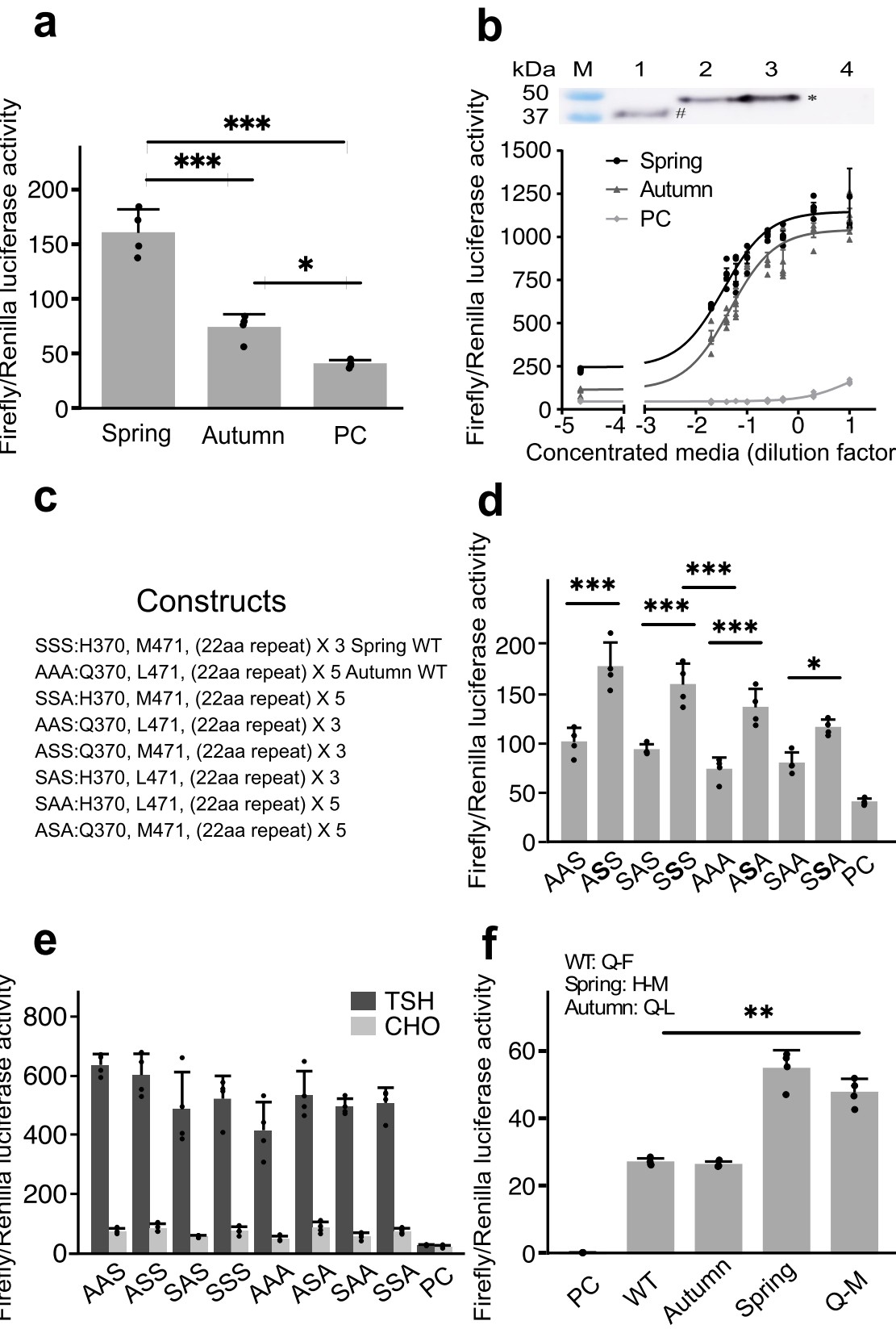

## Discussion

Previous population studies in Atlantic herring have documented that *TSHR* is clearly the most strongly differentiated locus between spring and autumn spawners in West-Atlantic, East-Atlantic and Baltic Sea populations[15,17,26,27]. A remarkable observation is that the same four SNPs, the two non-coding SNPs and the two missense mutations in *TSHR* highlighted in Fig. 2c, show very strong association in each geographic region[15,17]. Another two highly differentiated genetic variants, the 5.2 kb insertion/deletion of a BEL/Pao retrotransposon sequence and the *TSHR* C-terminal copy number variation, were added to the list of candidate causal mutations in the present study. The genotype/

**Fig. 5 Functional analysis of how herring *TSHR spring* and *autumn* alleles affect cAMP signaling. a** Comparison of constitutive activities of *spring* and *autumn* herring TSHRs expressed in EPC cells, monitored by a pGL4-CRE-dual-luciferase assay. PC: empty pcDNA3.1(+) plasmid. **b** Effects of recombinant herring scTSH on activating *spring* or *autumn* herring TSHR expressed in EPC cells. Top figure shows the western blot analysis of recombinant herring scTSH. Concentrated media containing herring scTSH incubated with PNGase F (lane 1) or without (lane 2) at 37 °C for 2 h, concentrated media from CHO-scTSH (lane 3) or untransfected CHO cells (lane 4) are tested in the western blot. M: molecular weight marker. Deglycosylated (#) and glycosylated (*) recombinant herring scTSH are indicated. Bottom figure shows the cAMP signaling activity induced by a serial dilution of the concentrated media containing herring TSH. X axis shows the log10-transformed dilution factors of the concentrated media. **c** All possible combinations of the three coding variants used in eight herring TSHR constructs. **d** Constitutive activities of EPC cells transfected with different herring TSHR constructs. **e** cAMP signaling comparisons among EPC cells transfected with different herring TSHR constructs after TSH induction for 4 h. **f** Constitutive activity assessed in HEK293 cells transfected with human TSHR WT or mutant constructs. Error bars represent the SDs calculated from four replicates ($N = 4$) at each data point. Unpaired two-tailed Student's *t* test was used for the statistical analysis. *: $P < 0.05$, **: $P < 0.01$, ***: $P < 0.001$.

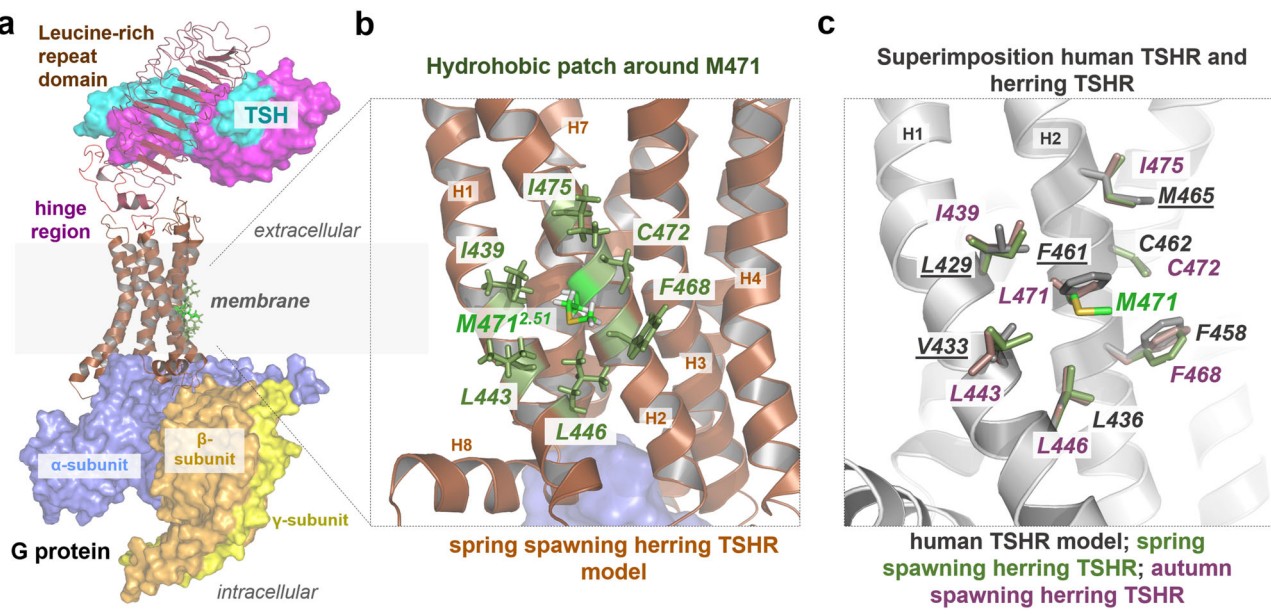

**Fig. 6 TSHR complex structural models and comparison of the TMH2 L471M variant. a** The entire complex model of herring TSHR with M471 highlighted in green (position 2.51) shows that this residue is outside the ligand or Gs-protein binding site. **b** Close-up view of M471[2.51] reveals that the side chain is directed towards the membrane and surrounded by a tight hydrophobic patch. **c** Superimposition of herring and human TSHRs reveals similarities and differences within the hydrophobic patch and also the potential side-chain orientations of leucine (autumn herrTSHR), methionine (spring herrTSHR), and phenylalanine (humTSHR). The human TSHR side chains that differ from those of herring TSHR are underlined.

phenotype relationship between *TSHR* polymorphisms and spawning time is very strong but it is not complete. The reason is that this is not a monogenic trait and there is a certain amount of plasticity which allows some gene flow between spring- and autumn-spawning herring. This explains the rare occurrence of the autumn *TSHR* allele in spring-spawning herring and vice versa (Fig. 2c). Despite the occurrence of gene flow, we find strong linkage disequilibrium in the region and very few recombinant haplotypes indicating that recombination within the *TSHR* region is suppressed and/or that recombinant haplotypes have reduced fitness.

Spring and autumn spawning are different reproductive strategies in the Atlantic herring. The disadvantage with autumn spawning is slow growth during the first period of life due to poor plankton production during winter. However, the advantage is that those that survive the winter are ahead of the spring-spawning larvae in development when plankton production starts during the spring. It is very likely that these different reproductive strategies have evolved over many generations in the Atlantic herring. We have previously reported that the *TSHR* haplotype blocks strongly associated with spring and autumn spawning are large, about 120 kb[17]. The expectation is therefore that these haplotype blocks harbor multiple sequence variants of functional importance that have accumulated during an evolutionary

process and thereby improved the fitness of the spring and autumn haplotypes. The present study provides conclusive evidence for the functional significance of the missense mutation L471M. The best candidate non-coding mutation is clearly the 5.2 kb insertion/deletion polymorphism involving a BEL/Pao retrotransposon located 2.15 kb upstream of the *TSHR* transcription start sites (Fig. 2b).

We show here using transfection experiments that the missense mutation L471M is associated with an enhanced constitutive activation of cAMP signaling. The functional significance of this variant is consistent with one of the major findings in the Atlantic herring genome project, namely that missense mutations have played a critical role for ecological adaptation in this species[15]. This amino acid position is located in the second transmembrane helix of TSHR. The spring TSHR allele (with methionine at this position) showed increased cAMP signaling activity, which was identified as a constitutively active mutant (so-called CAM: L471M), compared to both human (phenylalanine at this position) and the autumn variant of herring TSHRs (leucine at this position) (Fig. 5a). In the structural model, this position is located at a central site of a hydrophobic patch consisting of six amino acid residues between the helices TMH1 and TMH2 (Fig. 6b), which is involved in the stabilization and fixation of this structural part relative to the membrane. We assume that the longer and slimmer

methionine side chain leads to a lower fixation of the hydrophobic interactions compared to the bulkier and more hydrophobic leucine variant (autumn herring TSHR) or phenylalanine (human TSHR) and finally to an increased flexibility and mobility between the two helices, which might be associated with the observed higher permanent basal activity of this variant. An increase in basal activity by side-chain substitutions in vitro and in vivo is well known specifically for the human TSHR[28], whereby mutations of other hydrophobic residues in the transmembrane region oriented towards the membrane such as F631L in TMH6[29] or C672Y in TMH7[30] were already reported as CAMs. However, the identified CAM in our study, L471M substitution in TMH2, is remarkable, because all known (human TSHR) CAMs in TMH2, such as M453T[31] or M463V[32], are located inside the helical bundle between various helices. Their occurrence in human is often associated with congenital (non-autoimmune) hyperthyroidism.

How such an observed difference in basal signaling capacity affects the physiology and development of fishes is not known so far, due to the lack of animal models that simulate hyperactivity of TSHR. However, an enhanced TSHR signaling should promote higher DIO2 expression which in turn will lead to an increase in T3 levels and release of gonadotropin-releasing hormone that support gonadal maturation[10,33,34]. TSHR with higher constitutive activity, as observed for the variant associated with spring spawning, is expected to be less dependent on upstream input related to day length for signaling activity. Therefore, a possible consequence could be that herring carrying the TSHR spring variant show more plasticity in the timing of reproduction. Interestingly, data indicate in fact that it is more common that a herring born during spring spawning can take part in autumn-spawning than vice versa[35].

The presence of the 5.2 kb BEL/Pao retrotransposon located 2.15 kb upstream of the TSHR transcription start sites in the TSHR spring allele (Fig. 2b) may influence transcriptional regulation directly by introducing sequences with enhancer or promoter activity or indirectly by affecting the chromatin organization of the region. The latter possibility is supported by our discovery of a single robust open chromatin peak ~1.9 kb upstream of the BEL/Pao insertion, detected using ATAC-seq of brain and the part of brain containing saccus vasculosus, the structure assumed to play a critical role for photoperiodism in fish[12]. The region harboring the ATAC-seq peak also showed high sequence conservation among all 12 fish species used in the phastCons calculation (Fig. 3b). These results suggest that this region may represent an evolutionarily conserved cis-regulatory element. However, we have not yet been able to compare the temporal profile of TSHR expression for the spring and autumn alleles due to the challenges in sampling fish outside the spawning season of this pelagic fish and keeping them in captivity until sexual maturity.

We have been able to partially reconstruct the evolutionary history of herring TSHR haplotypes. We can conclude that the insertion of the BEL/Pao retrotransposon must have occurred prior to the split between Pacific and Atlantic herring about two million years before present[15]. The retrotransposon has subsequently recombined between its LTRs and lost internal sequence independently in the TSHR haplotype associated with autumn-spawning in Atlantic herring and in a TSHR haplotype present in Pacific herring. Loss of the BEL/Pao internal sequence in the autumn allele is a candidate causal mutation promoting autumn spawning because Pacific herring, in which the presence of intact BEL/Pao dominates, is primarily a spring spawner. It will therefore be of considerable interest to study the frequency of the Pacific Bel/Pao solo LTR haplotype in Pacific herring spawning at different time periods. Our analysis of nucleotide diversity (π) and Tajima's D indicates that both the autumn and spring TSHR haplotypes have gone through selective sweeps, and that the one

affecting the spring haplotype represents a more recent event (Fig. 1b–d). It is worth noticing that our comparison of Pacific and Atlantic herring shows that the spring allele M471 affecting TSHR cAMP signaling is the derived allele, in fact the spring allele carries the derived allele at 4 out of the 6 top sequence variants explored in this study (Fig. 2c). Thus, our data are consistent with an evolutionary scenario where some sequence variants have been selected because they promote autumn-spawning while others have been selected because they facilitate spring spawning resulting in selective sweep signals in both haplotype groups.

Insights gained from the in-depth functional analysis of the herring TSHR locus presented here contribute to the understanding of the molecular mechanisms how TSHR affects seasonal reproduction, a phenotype of considerable importance given the ongoing climate change that is affecting animal populations worldwide. The results may also have important implications in aquaculture since gene editing may be applied to manipulate the spawning season. Future analysis of this locus may involve in vivo experiments where the functional consequences of candidate mutations (e.g. missense mutations and BEL/Pao structural variant) are tested in transgenic model organisms exposed to different day lengths.

## Methods

**Animals**. Tissue samples for expression analysis were collected on August 24 and September 9 2016 from a population of spring-spawning herring kept in captivity at University of Bergen; the rearing of captive herring was approved by the Norwegian national animal ethics committee (Forsøksdyrutvalget FOTS ID-5072). The tissue samples used for ATAC-sequencing were collected at Hästskär on June 19 2019 from wild spring-spawning Atlantic herring in the Baltic Sea, which do not require ethical permission.

**Genome scan and genetic diversity at the TSHR locus**. We used a $2 \times 2$ contingency $X^2$ test to estimate the extent of SNP allele frequency differences between seven spring- and seven autumn-spawning herring populations from the Northeast Atlantic (Supplementary Table 2), and thus, identify the major genomic regions associated with seasonal reproduction. The SNP allele frequencies were generated in a previous study[16] and were derived from Pool-seq data. For the $X^2$ test, we formed two groups, spring and autumn spawners, and summed the reads supporting the reference and the alternative alleles for the pools included in each group.

To characterize genetic diversity at the TSHR locus, we calculated nucleotide diversity (π) and Tajima's D for the same seven spring- and seven autumn-spawning Atlantic herring populations used in the genome scan ($n = ~41–100$ per pool) (Supplementary Table 2). The whole-genome re-sequence data of these pools were previously reported by Han et al.[16] (for details of the pools used here see Supplementary Table 2). Unbiased nucleotide diversity π and Tajima's D were calculated for each pool using the program PoPoolation 1.2.2[36], which accounts for the truncated allele frequency spectrum of pooled data. In brief, a pileup file of chromosome 15 was generated from each pool BAM file using samtools v.1.10[37,38]. Indels and 5 bp around indels were removed to exclude spurious SNPs due to misalignments around indels. To minimize biases in the π and Tajima's D calculations, which are sensitive to sequencing errors and coverage fluctuations[39], the coverage of each pileup file was subsampled without replacement to a uniform value based on a per-pool coverage distribution (the target coverage corresponded to the 5th percentile of the coverage frequency distribution, which was used as the minimum coverage allowed for an SNP to be included in the analysis) (Supplementary Table 2). We also calculated the diversity parameters but skipping the coverage subsampling step and obtained very similar results with both approaches (Supplementary Fig. 5), thus, we decided to keep working with the subsampled datasets as coverage subsampling is recommended by the software developers[36]. To exclude spurious SNPs associated with repetitive sequences and copy number variants, we applied a maximum coverage filter equivalent to the per-pool 99th percentile of the coverage frequency distribution (Supplementary Table 2). A minimum base quality of 20, a minimum mapping quality of 20, and a minor allele count of 2 were required to retain high quality SNPs for further analysis. Both, π and Tajima's D statistics were calculated using a sliding window approach with a window size of 10 kb and a step size of 2 kb (the selected window-step combination offered a good genomic resolution while reducing the noise from single SNPs, after testing windows of 5, 10, 20, 40, 50, and 100 kb for non-overlapping and overlapping windows with a step size equivalent to 20% of the window size). Only windows with a coverage fraction ≥ 0.5 were included in the computations. In addition, we estimated the effective allele frequency difference, or delta allele frequency (dAF), between spring and autumn spawning groups at the

*TSHR* locus using the formula dAF = abs(mean(spring pools)−mean(autumn pools)). For each of the diversity parameters, we assessed whether the mean differences between sets of SNPs within chr 15: 8.85–8.95 Mb (215 SNPs) and outside (214 635 SNPs) the *TSHR* region were statistically significant among spring- and autumn-spawning groups using a Wilcoxon test. Data postprocessing, statistical tests, and plotting were performed in the R environment[40] (for specific parameters used in PoPoolation, see the associated code to this publication).

**Identification of the 5.2 kb structural variant.** Sequences spanning the entire *TSHR* gene plus 10 kb upstream and downstream from two reference assemblies, ASM96633v1[15] and Ch_v2.0.2[18], were aligned using BLAST[41] and the output was subsequently processed with a custom R script[40]. Repeats were annotated by CENSOR[21] for the region harboring the 5.2 kb structural variant. To validate the structural variant, long-range PCR was performed with genomic DNA from two spring- and autumn-spawning Atlantic herring in a 20 µL reaction containing 0.8 mM dNTPs, 0.3 µM each of the forward and reverse 5.2kb-confirm primers (Supplementary Table 1) and 0.75 U PrimeSTAR GXL DNA Polymerase (TaKaRa) following the program: 95 °C for 3 min, 35 cycles of 98 °C for 10 s, 58 °C for 20 s and 68 °C for 2 min 30 s, and a final extension of 10 min at 68 °C.

**ATAC-seq analysis.** BSH and brain without BSH were dissected from two spring-spawning herring caught in the Baltic Sea and transported to the lab on dry ice, then kept at −80 °C before nuclei isolation. ATAC-seq libraries were constructed according to the Omni-ATAC protocol with minor modifications[42]. Briefly, tissue was homogenized in 2 ml homogenization buffer (5 mM CaCl2, 3 mM Mg(Ac)₂, 10 mM Tris-HCl (pH = 7.8), 0.017 mM PMSF, 0.17 mM ß-mercaptoethanol, 320 mM Sucrose, 0.1 mM EDTA and 0.1% NP-40) with a Dounce homogenizer on ice. 400 µl suspension was transferred to a 2 ml tube for the density gradient centrifugation with different concentrations of Iodixanol solution. After centrifugation, a 200 µl fraction containing the nuclei band was collected, stained with Trypan blue and counted with a Countess II Automated Cell Counter (Thermo Fisher Scientific). An aliquot of 100,000–200,000 nuclei was used as input in a 50 µl transposition reaction containing 2 X TD buffer and 100 nM assembled Tn5 transposase for a 30-min incubation at 37 °C. Tagmented DNA was purified with a Zymo clean kit (Zymo Research). Purified DNA was used for an initial pre-amplification for 5 cycles, and the additional amplification cycle was determined by qPCR based on the "R vs Cycle Number" plot[43]. Amplified libraries were purified with a Zymo clean kit again, and library concentrations and qualities were evaluated using the 2200 TapeStation System (Agilent Technologies).

ATAC-seq was performed with a MiniSeq High Output Kit (150 cycles) on a MiniSeq instrument (Illumina) and 7–9 million reads were generated for each ATAC-seq library. Quality control, trimming, mapping, and peak calling of the sequenced reads were conducted following the ENCODE ATAC-seq pipeline (https://www.encodeproject.org/atac-seq/). The trimmed reads were aligned to the Atlantic herring reference genome (Ch_v2.0.2)[18] with Bowtie2[44] and the mapping rate was 85–95%. Duplicate reads, reads with low mapping quality and those aligned to the mitochondria genome were removed. The remaining reads (4–5 million) were subjected to peak calling by MACS2[45], where 22–32 K peaks were called. Sequenced library qualities were further evaluated by calculating the TSS enrichment score and checking the library complexity with the Non-Redundant Fraction (NRF) and PCR Bottlenecking Coefficients (PBC1 and 2). Finally, conserved peaks between two biological replicates were identified by evaluating the irreproducible discovery rate (IDR).

**Genotyping of six differentiated variants and haplotype analysis.** All six genetic variants, including the 5.2 kb structural variant, two non-coding SNPs, two missense SNPs and the copy number variant of C-terminal 22aa repeat, were genotyped in 45 spring-, 67 autumn-spawning Atlantic herring and 13 Pacific herring. TaqMan Custom SNP assays were performed to genotype the four SNPs in 5 µl reactions with a template of 20 ng genomic DNA (ThermoFisher Scientific). Copy number of the C-terminal 22aa repeat was determined by the PCR product size generated with geno22aa primers. Genotyping of the 5.2 kb structural variant was performed in a PCR reaction containing two forward primers (geno5.2kb-1F and geno5.2kb-2F) and one reverse primer (geno5.2kb-R), which generated PCR products with different sizes between spring and autumn spawners. All the primers used for genotyping are listed in Supplementary Table 1.

**Tissue expression profiles by quantitative PCR.** Total RNA was prepared from gonad, heart, spleen, kidney, gills, intestine, hypothalamus and saccus vasculosus (BSH), and brain without BSH (brain) of six adult spring-spawning Atlantic herring using RNeasy Mini Kit (Qiagen). RNA was then reverse transcribed into cDNA with a High-Capacity cDNA Reverse Transcription Kit (ThermoFisher Scientific). TaqMan Gene Expression assay (ThermoFisher Scientific) containing 0.3 µM primers and 0.25 µM TaqMan probe (Integrated DNA Technologies) was performed to compare the relative expression levels of *TSHR* among different tissues. qPCR with SYBR Green chemistry was used for *TSHB* and *DIO2* in a 10 µl reaction of SYBR Green PCR Master Mix (ThermoFisher Scientific) and 0.3 µM primers, with a program composed of an initial denaturation for 10 min at 95 °C followed by 40 cycles of 95 °C for 15 s and 60 °C for 1 min. Ct values were first

normalized to the housekeeping gene *ACTIN*, then the average expression for each gene in the gonad was assumed to be 1 for the subsequent calculation of the relative expression in other tissues.

**Plasmid constructs.** The coding sequence for the herring single-chain TSH (scTSH) was designed following a strategy previously used for mammalian gonadotropins[46] that contained an in-frame fusion of the cDNA sequences (5′–3′) of herring TSH beta subunit (NCBI: XM_012836756.1) and alpha subunit (NCBI: XM_012822755.1) linked by six histidines and then the C-terminal peptide of the hCG beta subunit. The designed sequence should generate a protein with a size of 30.6 kDa. Both scTSH and spring herring *TSHR* cDNA sequences were synthesized in vitro and cloned in the expression vector pcDNA3.1 by Genscript (Leiden, Netherlands). pcDNA3.1 plasmid expressing human *TSHR* was kindly provided by Drs. Gilbert Vassart and Sabine Costagliola (Université libre de Bruxelles, Belgium). Then, the spring herring *TSHR* and human *TSHR* plasmids were used as templates for site-directed mutagenesis to generate constructs coding for different mutant herring or human *TSHRs*. Plasmids for the dual-luciferase assay, including pGL4.29[luc2P/CRE/Hygro] containing cAMP response elements (CREs) to drive the transcription of luciferase gene *luc2P* and pRL-TK monitoring the transfection efficiency were purchased from Promega. Five ng of each plasmid was used to transform the XL1-Blue competent cells (Agilent), plasmid DNA was subsequently extracted from 200 ml overnight culture of a single transformant clone using an EndoFree plasmid Maxi Kit (Qiagen).

**Cell culture.** Chinese hamster ovary (CHO) (ATCC CCL-61) and human embryonic kidney 293 (HEK293) (ATCC CRL-1573) cells were maintained in DMEM supplemented with 5% (CHO) or 10% FBS (HEK293), 100 U/ml penicillin, 100 µg/ml streptomycin and 292 µg/ml L-Glutamine (ThermoFisher Scientific) at 37 °C with 5% CO₂. Epithelioma Papulosum Cyprini (EPC) cells (ATCC CRL-2872) were cultured in EMEM (Sigma) supplemented with 10% FBS, 100 U/ml penicillin, 100 µg/ml streptomycin, 292 µg/ml L-Glutamine and 1 mM Sodium Pyruvate (ThermoFisher Scientific) at 26 °C with 5% CO₂.

**Production of recombinant herring scTSH.** CHO cells were transfected with the scTSH expression plasmid using Lipofectamine 3000 (Invitrogen), stable clones were subsequently selected with 500 µg/ml G418 (Invitrogen) and screened for producing scTSH by western blot using a polyclonal antisera against the sea bass alpha subunit[47]. A positive clone was expanded in 225 cm² cell culture flasks (Corning) in culture medium containing 5% FBS until confluence, then the cells were maintained in serum-free DMEM for hormone production for 7 days at 25 °C[48]. After 7 days, culture medium containing scTSH or without (negative control) was centrifuged at 15000 x g for 15 min and concentrated by ultrafiltration using Centricon Plus-70 / Ultracel PL-30 (Merck Millipore Ltd.). Then, western blotting was performed to confirm TSH production. Concentrated medium containing herring scTSH was denatured at 94 °C for 5 min in 0.1% SDS and 50 mM 2-mercaptoethanol, and then treated with 2.5 units of peptide-N-glycosidase F (Roche Diagnostics) at 37 °C for 2 h in 20 mM sodium phosphate with 0.5% Nonidet P-40, pH 7.5. All samples were run in 12% SDS-PAGE in the reducing condition and transferred to a PVDF membrane (Immobilon P; Millipore Corp.), then blocked overnight with 5% skimmed milk at 4 °C. After blocking, the membrane was incubated with polyclonal antisera against the sea bass alpha subunit (dilution 1:2000) for 90 min at room temperature, washed, and then further incubated with 1:25000 goat anti-rabbit immunoglobulin G (IgG) horseradish peroxidase conjugate (Bio-Rad Laboratories) for 60 min at room temperature. Immunodetection was performed by chemiluminescence with a Pierce ECL Plus Western Blotting Substrate kit (ThermoFisher Scientific).

**Cell surface expression.** A Rhotag (MNGTEGPNFYVPFSNKTGVVYEE) was inserted at the N-terminus of herring TSHR for flow cytometry analysis of receptor cell surface expression. Anti-Rhotag polyclonal antibody was kindly provided by Drs. Gilbert Vassart and Sabine Costagliola (Université Libre de Bruxelles, Brussels, Belgium). PBS containing 1% BSA and 0.05% sodium azide was prepared as the flow cytometry (FCM) buffer for the washing and antibody incubation steps. 2.2 × 10⁶ EPC cells were seeded in a 100 mm poly-D-Lysine-treated petri dish the day before transfection. Each dish was transfected with 10 µg TSHR or empty pcDNA3.1 expression plasmid using 20 µl jetPRIME transfection reagent in 500 µl jetPRIME transfection buffer (Polyplus transfection). Cells were harvested 24 h after transfection, then washed once in cold PBS and fixed in 2% PFA for 10 min at room temperature. After fixation, cells were washed three times with FCM buffer, then incubated with anti-Rhotag antibody or FCM buffer (negative control) for 1 h at room temperature. Cells were washed again with FCM buffer three times and stained with Alexa Fluor 488-labeled chicken anti-mouse IgG (H + L) antibody (1:200 dilution, ThermoFisher Scientific) or FCM buffer (negative control) for 45 min in the dark. After the fluorescent staining, cells were washed three times and resuspended in FCM buffer before analysis on a CytoFLEX instrument (Beckman Coulter). A minimum of 100,000 events was recorded for each sample, fluorescence intensities of negative control and cells transfected with empty pcDNA3.1 plasmid were used as the background for gating strategy. Cell surface expression was

represented by the mean fluorescence intensity of the positively stained cell population.

**Dual-luciferase reporter assay**. EPC or HEK293 cells were plated in a 48-well plate at a density of $1 \times 10^5$ cells/well the day before transfection. A total of 250 ng plasmid mixture containing pGL4.29[luc2P/CRE/Hygro], TSHR expression plasmid (or empty pcDNA3.1) and pRL-TK with the ratio of 20:5:1 was prepared to transfect each well of cells using jetPRIME transfection reagent (Polyplus). Medium was replaced by fresh medium containing 10% FBS (TSH-induced condition) or serum-free medium (constitutive activity condition) 4 h after transfection. On day three, cells were treated with serum-free medium containing different dilutions of the concentrated scTSH medium for 4 h (TSH-induced condition) or directly subjected to the luminescence measurement without TSH induction (constitutive activity condition). Luminescence was measured using a Dual-Luciferase Reporter assay (Promega) on an Infinite M200 Microplate Reader (Tecan Group Ltd., Switzerland), and luciferase activity was represented as the ratio of firefly (pGL4.29 [luc2P/CRE/Hygro]) to Renilla (pRL-TK) luminescence.

**5′-RACE to identify the herring *DIO2* TSS**. Total RNA was prepared from brain of a spring-spawning Atlantic herring using the RNeasy Mini Kit (Qiagen). Six μg of the isolated RNA was used for 5′-RACE with a FirstChoiceTM RLM-RACE Kit (ThermoFisher Scientific). One μl cDNA or Outer RACE PCR product was used as PCR template in a 20 μL reaction containing 0.8 mM dNTPs, 0.3 μM of each forward and reverse primer (Supplementary Table 1) and 0.75 U PrimeSTAR GXL DNA Polymerase (TaKaRa). Amplification was carried out with an initial denaturation of 3 min at 95 °C, followed by 35 cycles of 98 °C for 10 s, 58 °C for 20 s and 68 °C for 40 s, and a final extension of 10 min at 68 °C. The final 5′ RACE product was sequenced at Eurofins Genomics (Ebersberg, Germany).

**Sequence conservation analysis**. Genomic sequences covering the *TSHR* locus were extracted from Ensembl Genome Browser for Atlantic herring and 11 other fish species, including Amazon molly (*Poecilia formosa*), denticle herring (*Denticeps clupeoides*), goldfish (*Carassius auratus*), guppy (*Poecilia reticulata*), *Neolamprologus brichardi*, Japanese medaka (*Oryzias latipes*), northern pike (*Esox lucius*), orange clownfish (*Amphiprion percula*), spotted gar (*Lepisosteus oculatus*), three-spined stickleback (*Gasterosteus aculeatus*) and spotted green pufferfish (*Tetraodon nigroviridis*). The extracted sequences were firstly aligned using progressiveCactus[49,50], and a subsequent alignment was generated using the hal2maf program from halTools[51] with Atlantic herring assembly (Ch_v2.0.2)[18] as the co-ordinate backbone. This alignment was used for the downstream phastCons score calculation by running phyloFit[24] and phastCons[25] from the PHAST package with default parameters. Peaks were called by grouping signals with a minimum phastCons score of 0.2 within 500 bp region.

**Structure modeling of human and herring TSHRs**. In order to explore the possible interactions of the variant residues with other receptor interacting proteins and to study intramolecular interactions, we built a structural homology model for the herring TSHR (herrTSHR) complexed with herring TSH and Gs-protein. The TSHR hinge region that harbors the Q370H substitution and the C-terminus containing the 22aa repeat were excluded from the homology model due to the lack of structural templates for these regions. The homology model was constructed by using the following structural templates of evolutionarily related class A GPCRs: (i) the leucine-rich repeat domain (LRRD) complexed with hormone was modeled based on the solved FSHR LRRD - FSH complex structure (Protein Data Bank (PDB) ID: 4AY9)[52,53], this part of model included herring TSHR Cys33 - Asn296 and fragments of the hinge region Gln297 - Thr312 and Ser393 - Ile421; (ii) the available structural complex of β2-adrenoreceptor with Gs-protein (PDB ID: 3SN6)[54] was used as the template to model the seven-transmembrane helix domain (7TMD) of herring TSHR in the active conformation; (iii) the extracellular loop 2 (ECL2) was built by using the ECL2 of μ-opioid receptor (PDB ID: 6DDE)[55]. To prepare the template for herring TSHR modeling, the fused T4-lysozyme and bound ligand of β2-adrenoreceptor were deleted, the ECL1 and ECL3 loops were adjusted manually to the loop length of herring TSHR. Due to the lack of third intracellular loop (ICL3) in the β2-adrenoreceptor structure, amino acid residues of herring TSHR ICL3 were manually added to the template. Since herring TSHR does not have the TMH5 proline, which is highly conserved among all class A GPCRs and responsible for the helical kinks and bulges within this region[56], we assumed a rather regular (stretched) helix conformation for the herring TSHR TMH5 and therefore replaced the kinked β2-adrenoreceptor TMH5 template with a regular α-helix. Moreover, the ECL2 template was substituted with μOR ECL2 structure because of its higher sequence similarity with herring TSHR in this region. Finally, amino acid residues of this chimeric 7TMD template and FSHR N-terminus were mutated to the corresponding spring herring TSHR residues and sequence of the heterodimeric FSH ligand was substituted by the herring TSH. All homology models were generated by using SYBYL-X 2.0 (Certara, NJ, US). The 7TMD structure was then fused with FSHR N-terminus at position 421. The assembled complex was subsequently optimized by the energy minimization under constrained backbone atoms (the AMBER F99 force field was used), followed by a

2 ns molecular dynamics simulation (MD) of the side chains. The entire TSHR complex was energetically minimized without any constraints until converging at a termination gradient of 0.05 kcal/mol*Å. Next, for autumn herrTSHR modeling, the spring TSHR sequence was substituted with autumn TSHR sequence. For humTSHR, the spring TSHR sequence was substituted with human TSHR, and the herring TSH ligand was replaced by the bovine TSH sequence. Both complex models were energetically minimized until converging at a termination gradient of 0.05 kcal/mol*Å.

To investigate the microenvironment around the L471M mutation at TMH2 position 2.51, local short MD's of 4 ns on Met471[2.51] (spring herrTSHR), Leu471 (autumn herrTSHR) or Phe461 (humTSHR) and its surrounding amino acids were performed. During MD simulations, backbone atoms of the entire complexes as well as all side chains, except residues at positions 1.47, 1.51, 1.54, 2.48, 2.52, and 2.55 that form the hydrophobic patch around position 2.51, were constrained.

**Statistics and reproducibility**. Results were presented as the mean + SD (standard deviation) calculated from at least four biological replicates for each experiment, and at least two independent experiments were conducted for each assay. Unpaired two-tailed Student's *t* test was performed to calculate the P-values and means were judged as statistically significant when $P \leq 0.05$.

**Reporting summary**. Further information on research design is available in the Nature Research Reporting Summary linked to this article.

## Data availability

The raw ATAC-seq reads have been deposited in the NCBI Sequence Read Archive (SRA) under BioProject accession no. PRJNA660525. The SNP allele frequency data from Han et al.[16] used to perform the genome scan is available in Dryad (https://doi.org/10.5061/dryad.pnvx0k6kr)[57], and the corresponding sequence data is available in NCBI under the Bioproject PRJNA642736. The raw data to build Fig. 1c has been uploaded to the GitHub repository as a compressed directory https://github.com/LeifAnderssonLab/Chen_et_al_2021_TSHR_functional/blob/main/data-pi-tajimasD-popoolation1.tar.gz, DOI: 10.5281/zenodo.4733728[58]. All other raw data supporting the findings of this study are provided in Supplementary Data 1.

## Code availability

Custom code and associated data files used for the genome analysis are available in the repository https://github.com/LeifAnderssonLab/Chen_et_al_2021_TSHR_functional, https://doi.org/10.5281/zenodo.4733728[58].

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

## Acknowledgements

Sincere thanks are due to Gilbert Vassart and Takashi Yoshimura for valuable advice on the study, Mårten Larsson for TSHR construct design, Arild Folkvord for access to herring samples, Arianna Cocco for collection of tissue samples, and Soledad Ibañez for technical support for recombinant herring TSH production. We also thank Gilbert Vassart and Sabine Costagliola for providing the human *TSHR* expression plasmid and the anti-Rhotag antibody for flow cytometry. This work was funded by Vetenskapsrådet (to L.A. and P.J.), the Knut and Alice Wallenberg foundation (to L.A.), Research Council of Norway project 254774, GENSINC (to L.A.) and Formas (to P.J.). Chunheng Mo was supported by the Overseas Exchange Program for Ph.D Students from China Scholarship Council (No. 201706240029). P.S. acknowledges funding by the Deutsche For-schungsgemeinschaft (DFG, German Research Foundation) through CRC 1423, project number 421152132, subprojects A01, A05, Z03; through CRC 1365, project number 394046635, subproject A03 (to G.K. and P.S.); through Germany's Excellence Strategies —EXC 2008/1 (UniSysCat)—390540038. We thank the SNP&SEQ Technology Platform and UPPMAX in Uppsala for the provided service in high-throughput sequencing and computational infrastructure under the SNIC project 2020/15-137. These facilities are part of the National Genomics Infrastructure (NGI) of Sweden and Science for Life Laboratory. The SNP&SEQ Platform is also supported by the Swedish Research Council and the Knut and Alice Wallenberg Foundation.

## Author contributions

L.A. and J.C. conceived the project and designed the study. J.C. identified the BEL/Pao structural variant, performed haplotype analysis, transfection experiments, and tissue expressions. M.P. analyzed sequence conservation and heterozygosity. D.X.S. carried out ATAC-seq. S.Y. contributed to the expression analysis. A.P.F.P. performed the genome scan and the genome-wide diversity analysis. H.B. and C.M. contributed to the design and performance of luciferase assays. P.J. contributed to the characterization of BEL/Pao structural variant. G.M. and A.G. produced the recombinant herring scTSH. G.K. and P.S. modeled the TSHR/TSH/Gs-protein complex and participated in the design of mutagenesis analysis. J.C. and L.A. wrote the manuscript with contributions from other authors. All authors approved the manuscript before submission.

## Funding

## Competing interests

The authors declare no competing interests.
