## [Peer Review File · Communications Biology]

Reviewers' comments:

Reviewer #1 (Remarks to the Author):

The study by Chen and colleagues builds on previously published evidence of genomic regions strongly associated to variation in spawning season in Atlantic herring, by looking in detail in one of those regions—the one containing the most significant associated peak—to try to identify the causal mutation using a battery of molecular techniques. The study is highly interesting and for the most part well design, presenting state of the art molecular techniques to identify the functional consequences of genetic polymorphisms. The manuscript is easy to read and to follow, and the figures are clear. Overall, I enjoyed reading this manuscript. I have two main concerns and a few minor comments that I describe below.

1. The authors report a selective sweep based on the decay of heterozygosity in the THSR region. There is no test associated with this, and figure 1 is not completely clear. For example, there seem to be no markers in the middle of the THSR region. It is unclear to me if there are no markers in this region or if those are not polymorphic within morph (i.e., spring versus autumn spawners). Also, the decay of heterozygosity is not that clear, especially for autumn spawners if it is only based on Fig. 1C. As the presence of the selective sweep is the base of some of the discussion, it would be useful if the authors could clarify this further.

2. The authors come to the conclusion that two mutation at the THSR locus might have functional consequences for the season of spawning. One of such mutations is in the coding region of the THSR and the authors show that it enhances constitutive activity of the THS-receptor (although they found no increased THS-induced cAMP signaling activity, Fig. 5e). They conclude that this enhanced constitutive activity might increase cAMP affecting the expression of DIO2, which they show to have two cAMP response elements upstream of the DIO2 transcription start site. However, the authors provide no evidence that indeed DIO2 is differentially expressed between autumn and spring spawners. This is surprising as it is clear that the authors can effectively measure DIO2 expression (see fig. 4c). There is a similar scenario for the second mutation, an insertion of a retrotransposon element between an open chromatin region identified using ATAC-seq—putatively involved in the regulation of THSR expression— and the transcription start site for THSR (THSR TSS). This 5.2 kb insertion significantly increases the distance between the putative regulatory region and THSR TSS. This is suggested to affect the expression of THSR potentially influencing the spawning season. However, there is no evidence that THSR is differentially expressed between autumn and spring spawners. As the authors can quantify expression of this gene, it would be important to present this data to make the conclusions stronger. I imagine that one reason why these data are not presented, is because it might be complicated to determine the timing at which each of the two morphs need to be sampled to effectively capture differences in expression of these two genes. However, I imagine that a temporal profile showing seasonal variation of these genes in the two morphs might be very informative.

Below I present a series of minor comments (some might be repetitive with the main points above).

Best wishes!

L47: Maybe move "to seasonal changes" to the end of the sentence.

L49: I would rather use "e.g.," than "i.e." here.

L56: is this specific to mammals or to all vertebrates (as suggested later and by Nishiwaki-Ohkawa & Yoshimura 2016)? I see. The light pathway is specific to mammals. Anyway, the emphasis in mammals is unneeded and confusing as the work is in fish.

L70-72: clarity of this sentence can be improved.

L88-91: Are these mutagenesis analyses part of the results of this study or a previous study? There is no reference associated. If these are part of the results, please make it more explicit.

L180: I recommend moving the Haplotype analysis before the presentation of the ATAC-seq experiment, as the Haplotype analysis is still describing variants, and the ATAC-seq is moving to mechanisms and will link better to the expression data.

L227: I might be reading the data incorrectly here, but here a 1.2 fold increase is reported, but Fig 5a shows a ~2.2 fold increase in Spring compared to Autumn spawners (around 160 to 75

luciferase activity, respectively). Am I missing something?

L240: "signaling involves in the regulation" -> "signaling is involved in the regulation"

L298 & L326-328: the functional importance of the BEL/Pao structural variant is inferred solely from the increased distance between the THSR gene and the putative regulatory region, no? But no evidence that this is indeed the case is presented, no? I mean, is THSR differentially expressed between autumn and spring spawners?

L313: I think it is very interesting that most heterozygotes appear to be F1s, as they are heterozygotes for the six identified variants (with the only exception of the potentially younger H370Q. Is it possible that these "hybrids" have lower breeding success and therefore recombinants are not commonly seen?

L100-101: In Fig. 1b and c, it is not clear if in the center of the TSHR locus there are no heterozygotes or if there are no marker at all (which could well be just because there are no polymorphic sites in that range). Hence, it is hard to visually evaluate if there is or not a decrease in heterozygosity without any further information, especially for autumn-spawners. Basically, were all polymorphic sites in the region analyzed (I assume not based on Fig. 3)?

L204, L475, & Fig. 4: I have some questions with regard to the expression data. First, were autumn spawners or spring spawners used? Based on the Haplotype analysis and the ATAC-seq analysis it is concluded that the regulation of THSR might be affected; yet no information on the phenotype of the studied individuals is presented. Second, it is stated that it is relative expression. Is it relative to a housekeeping gene? or relative to expression in gonads (after correcting for HKG)? Last, why there is no comparison between autumn spawners and spring spawners?

L243: This is still indirectly inferred. Is there evidence that there is a change in DIO2 expression as expected for the difference in cAMP signaling activity between spring and autumn spawners?

L356-358: Given the impressive molecular work already presented in this manuscript, it is difficult to understand why expression of DIO2 is not compared between the two type of spawners. Data on the expression of this gene will make the conclusions about the functional importance of the L471M substitution less speculative. The authors are capable of measuring the expression of DIO2 as evidenced in Fig. 2c.

L365-374: This conclusion is as well confusing to me. It suggests that a retrotransposon insertion might be affected the expression of TSHR between the two morphs. But no evidence is presented for differential expression of TSHR between the autumn and spring spawners. This is strange as the authors are obviously capable of measuring the expression of this gene (see Fig. 2a). Thus, I think that this extra step will strongly reinforce the conclusion.

L384: This statement is at odds with that in line 200-202 saying "We have no phenotype data as regards spawning time for the Pacific herring included in this study as they were caught outside the spawning season."

L400: I would say that the study identified potential candidates, rather than the two identified are likely to contribute to this adaptation. The association is strong, but there are many steps missing to really determine the genotype/phenotype map.

L858-860: panel d, supposedly showing the expression of opsin genes, is missing from the figure.

Reviewer #2 (Remarks to the Author):

Dear editor,

Even though I'm used to review many articles per year, I must say that I'm not a specialist of the specific question asked within the present study. Also, I'm not able to evaluate the entire methods used. Nevertheless, I can give a broad overview of the article from a "naive" perspective; it is therefore very important that the MS must be evaluated by specialists able to assess the entire article.

Overall, I think that the question asked is relevant and interesting. I'm working on fish reproduction for the past 15 years, and for sure timing of reproduction is an important issue. The authors chose a fish species displaying both spring and autumn spawning, the Atlantic herring. Indeed, most temperate fish species spawn once a year (not the case for tropical fish species). I was wondering whether the authors look at rainbow trout (*Oncorhynchus mykiss*), because these species also spawn at different period of time (some domesticated and selected strains); I'm

unsure whether natural populations are able to do so, but I would say yes. I do not know whether similar works were performed on that species, but could be interesting to look at either for the present work or further research? Line 50; there are several articles for fish that clearly stated/demonstrated this, you may add Wootton, R. J. (1999). *Ecology of Teleost Fishes*, 2nd edn. Dordrecht: Kluwer Academic Publishers. ; Teletchea F, Fontaine P (2010) Comparison of early life-stage strategies in temperate freshwater fish species: trade-offs are directed towards first feeding of larvae in spring and early summer. *Journal of Fish Biology* (2010) 77, 257–278. In the results, how do you determine “a clear loss of heterozygosity” on the fig 1 (line 101). It is only visually. For me it is indeed quite “obvious” for spring spawning but much less for autumn spawners? The results are very clear and dense (the quantity and quality of data produced is impressive). The discussion is also very clear. I suggest, but you may not take into account this remark (of for further studies), to look at the aquaculture field because what you are looking at could be very interesting for this community as well, notably how the spawning season is “genetically determinate” and how it could be modified (and therefore how the genomic regions might be impacted?). The materials and methods are clear; yet I’m not enough competent to evaluate in detail what it is indicated. Overall, I consider that the MS is suitable for publication.

Minor comments

L70: please take a look at this sentence, might need to be rewritten?

L156: please add the scientific name of Pacific herring

L199: this suggests

L865: this is unclear to what this belongs “d) Relative expression of herring opsin genes in BR and BSH. Y axis 866 shows gene expression values as log₂-transformed TMM-normalized counts-per-million 867 (CPM). A total of 6-7 samples were included in each tissue group (N = 14).” There are no “d” in that Figure ?

L864: I propose to replace 1 by 100 because the Y-axis is from 0 to 150. It is unclear how I can read this figure, for instance F4A, this imply that the expression level in BSH is 20 fold the expression in gonad? Sorry I’m not very used to this kind of presentation. Might be a stupid remark.

L231: you can add the scientific name of sea bass

Response to the comments from Reviewers

We thank the Reviewers for their constructive criticism of our paper that has helped us to improve the paper. Here is a point-by-point response, in blue, to the reviewers' comments and concerns. All the line numbers refer to the revised manuscript file.

Reviewer #1 (Remarks to the Author):

The study by Chen and colleagues builds on previously published evidence of genomic regions strongly associated to variation in spawning season in Atlantic herring, by looking in detail in one of those regions—the one containing the most significant associated peak—to try to identify the causal mutation using a battery of molecular techniques. The study is highly interesting and for the most part well design, presenting state of the art molecular techniques to identify the functional consequences of genetic polymorphisms. The manuscript is easy to read and to follow, and the figures are clear. Overall, I enjoyed reading this manuscript. I have two main concerns and a few minor comments that I describe below.

Response: Thank you for the positive comments.

1. The authors report a selective sweep based on the decay of heterozygosity in the THSR region. There is no test associated with this, and figure 1 is not completely clear. For example, there seem to be no markers in the middle of the THSR region. It is unclear to me if there are no markers in this region or if those are not polymorphic within morph (i.e., spring versus autumn spawners). Also, the decay of heterozygosity is not that clear, especially for autumn spawners if it is only based on Fig. 1C. As the presence of the selective sweep is the base of some of the discussion, it would be useful if the authors could clarify this further.

Response: We agree that this was not crystal clear in the previous version and the gap in the middle occurred because there was a gap in our old assembly that was used to design the SNP array used to generate the data. We have therefore replaced this analysis with data based on individual whole genome sequence data that demonstrate the sweep signal much more strikingly and the drop in heterozygosity is strongly supported by statistical analysis.

2. The authors come to the conclusion that two mutation at the THSR locus might have functional consequences for the season of spawning. One of such mutations is in the coding region of the THSR and the authors show that it enhances constitutive activity of the THS-receptor (although they found no increased THS-induced cAMP signaling activity, Fig. 5e). They conclude that this enhanced constitutive activity might increase cAMP affecting the expression of DIO2, which they show to have two cAMP response elements upstream of the DIO2 transcription start site. However, the authors provide no evidence that indeed DIO2 is differentially expressed between autumn and spring spawners. This is surprising as it is clear that the authors can effectively measure DIO2 expression (see fig. 4c). There is a similar scenario for the second mutation, an insertion of a retrotransposon element between an open chromatin region identified using ATAC-seq—putatively involved in the regulation of THSR expression— and the transcription start site for THSR (THSR TSS). This 5.2 kb insertion significantly increases the distance between the putative regulatory region and THSR TSS. This is suggested to affect the expression of THSR potentially influencing the spawning season. However, there is no evidence that THSR is differentially expressed between autumn and spring spawners. As the authors can quantify expression of this gene, it would be important to present this data to make the conclusions stronger. I imagine that one reason

why these data are not presented, is because it might be complicated to determine the timing at which each of the two morphs need to be sampled to effectively capture differences in expression of these two genes. However, I imagine that a temporal profile showing seasonal variation of these genes in the two morphs might be very informative.

Response: The reviewer has raised an important point here which is unfortunately also a limitation of studying this pelagic species. We have sampled herring when they come for spawning at the coast when we can also determine their phenotype as spring- or autumn spawner. But it is much more difficult to get samples outside the spawning season and the fishery biologist we interact with cannot tell where we could sample spring- and autumn spawners separately. So, we have not yet been able to collect samples for a temporal study of gene expression. We agree with the reviewer that our conclusion would be stronger if we could compare the expressions of *TSHR* and *DIO2* between spring and autumn spawners and perform a temporal study, but we lack the samples to add this information unfortunately.

We have tried to explore the *TSHR* temporal expressions using spring spawning herring in captivity. Due to the difficulty in keeping pelagic fishes in the laboratory, we were only able to check the *TSHR* expressions at two pre-spawning and one post-spawning stage. We did observe an enhanced *TSHR* expression at the second pre-spawning stage compared to the first pre-spawning stage accompanying an increase in the GSI (unpublished data). Unfortunately, these fishes didn't spawn as expected and gonads regressed before reaching maturation, which indicated that spawning in herring is a complicated process and highly context-dependent. As the reviewer suggested, it's of great significance to investigate the temporal expression profiles of genes involved in the photoperiodic regulation of seasonal reproduction, we will explore other sampling possibilities for this analysis in our future studies.

1, L47: Maybe move "to seasonal changes" to the end of the sentence.

Response: We agree with this comment and have changed the text accordingly.

2, L49: I would rather use "e.g.," than "i.e." here.

Response: We agree with this comment and have changed "i.e." to "e.g."

3, L56: is this specific to mammals or to all vertebrates (as suggested later and by Nishiwaki-Ohkawa & Yoshimura 2016)? I see. The light pathway is specific to mammals. Anyway, the emphasis in mammals is unneeded and confusing as the work is in fish.

Response: Since the photoperiodic signaling pathway is well-studied in mammals and less investigated in fishes, we used the mammals here as a model to clarify the functions of key genes involved in this signal transduction pathway. We modified the sentence by stating that the mechanism underlying photoperiodism has been explored in mammals.

4, L70-72: clarity of this sentence can be improved.

Response: Thank you for the suggestion, we have rewritten the sentence on line 75.

5, L88-91: Are these mutagenesis analyses part of the results of this study or a previous study? There is no reference associated. If these are part of the results, please make it more explicit.

Response: We have revised this text to make it clearer what is previous results and what is presented in the current study. We now start a new paragraph when we make a short introduction to the results of the current study.

6, L180: I recommend moving the Haplotype analysis before the presentation of the ATAC-seq experiment, as the Haplotype analysis is still describing variants, and the ATAC-seq is moving to mechanisms and will link better to the expression data.

Response: Thank you for the suggestion, we have moved the “Haplotype analysis” section before the “ATAC-seq identifies an open chromatin region close to the 5.2 kb structural variant” section.

7, L227: I might be reading the data incorrectly here, but here a 1.2 fold increase is reported, but Fig 5a shows a ~2.2 fold increase in Spring compared to Autumn spawners (around 160 to 75 luciferase activity, respectively). Am I missing something?

Response: The reviewer is correct, we have changed “1.2-fold” to “2.2-fold” on line 290 in the revised manuscript.

8, L240: "signaling involves in the regulation" -> "signaling is involved in the regulation"

Response: We agree with this comment and have made the suggested changes on line 304 in the revised manuscript.

9, L298 & L326-328: the functional importance of the BEL/Pao structural variant is inferred solely from the increased distance between the THSR gene and the putative regulatory region, no? But no evidence that this is indeed the case is presented, no? I mean, is THSR differentially expressed between autumn and spring spawners?

Response: This is correct and we have added the following statement on line 442-445 to explain that differences in expression patterns have not been documented:

“However, we have not yet been able to compare the temporal profile of *TSHR* expression for the spring and autumn alleles due to the challenges in sampling fish outside the spawning season of this pelagic fish and in keeping herring in captivity until they reach sexual maturity.”

10, L313: I think it is very interesting that most heterozygotes appear to be F1s, as they are heterozygotes for the six identified variants (with the only exception of the potentially younger H370Q. Is it possible that these "hybrids" have lower breeding success and therefore recombinants are not commonly seen?

Response: We agree that this is interesting and most likely reflect the strong selection pressure on this locus. We have added the following sentence on line 379-382:

“Despite the occurrence of gene flow, we find strong linkage disequilibrium in the region and very few recombinant haplotypes indicating that recombination within the *TSHR* region is suppressed and/or that recombinant haplotypes have reduced fitness.”

11, L100-101: In Fig. 1b and c, it is not clear if in the center of the *TSHR* locus there are no heterozygotes or if there are no marker at all (which could well be just because there are no polymorphic sites in that range). Hence, it is hard to visually evaluate if there is or not a decrease in heterozygosity without any further information, especially for autumn-spawners. Basically, were all polymorphic sites in the region analyzed (I assume not based on Fig. 3)?

Response: As explained we have now replaced this analysis with an analysis of heterozygosity in individual whole genome sequence data, so this is clearer now.

12, L204, L475, & Fig. 4: I have some questions with regard to the expression data. First, were autumn spawners or spring spawners used? Based on the Haplotype analysis and the ATAC-seq analysis it is concluded that the regulation of *THSR* might be affected; yet no

information on the phenotype of the studied individuals is presented. Second, it is stated that it is relative expression. Is it relative to a housekeeping gene? or relative to expression in gonads (after correcting for HKG)? Last, why there is no comparison between autumn spawners and spring spawners?

Response: All the fishes used in this experiment were spring spawners. As regards the tissue profiling analysis, Ct value was first normalized to the housekeeping gene *ACTIN*, then the average expression for each gene in the gonad was assumed to be 1 for the subsequent calculation of the relative expression in other tissues. We have added both the sample information and relative expression analysis method in the method section “Tissue expression profiles by quantitative PCR” on pages 21.

We agree with the reviewer that it would be more informative if we could compare the gene expressions between spring and autumn spawners, but it’s very challenging to obtain the right samples, we have explained the reasons in the response to the major comment 2.

13, L243: This is still indirectly inferred. Is there evidence that there is a change in DIO2 expression as expected for the difference in cAMP signaling activity between spring and autumn spawners?

Response: Although we agree that this is an important consideration, but it’s difficult to obtain the right samples for this analysis, we have explained the reasons in the response to the major comment 2.

14, L356-358: Given the impressive molecular work already presented in this manuscript, it is difficult to understand why expression of DIO2 is not compared between the two type of spawners. Data on the expression of this gene will make the conclusions about the functional importance of the L471M substitution less speculative. The authors are capable of measuring the expression of DIO2 as evidenced in Fig. 2c.

Response: We agree with the reviewer that it would be more informative if we could compare the expressions of *DIO2* between spring and autumn spawners. However, obtaining the right samples is very challenging, we have explained the reasons in the response to the major comment 2.

15, L365-374: This conclusion is as well confusing to me. It suggests that a retrotransposon insertion might be affected the expression of TSHR between the two morphs. But no evidence is presented for differential expression of TSHR between the autumn and spring spawners. This is strange as the authors are obviously capable of measuring the expression of this gene (see Fig. 2a). Thus, I think that this extra step will strongly reinforce the conclusion.

Response: We agree with the reviewer that it would be more informative if we could compare the TSHR expressions between spring and autumn spawners, but it’s not easy to obtain the right samples, we have explained the reasons in the response to the major comment 2.

16, L384: This statement is at odds with that in line 200-202 saying "We have no phenotype data as regards spawning time for the Pacific herring included in this study as they were caught outside the spawning season."

Response: We have added the following text to line 216 (former 202) to make the two sentences consistent:

“, but Pacific herring is primarily spring spawning.”

17, L400: I would say that the study identified potential candidates, rather than the two identified are likely to contribute to this adaptation. The association is strong, but there are many steps missing to really determine the genotype/phenotype map.

Response: We have deleted this sentence since it is not essential.

18, L858-860: panel d, supposedly showing the expression of opsin genes, is missing from the figure.

Response: Since we think these opsin expression data is not much related to our *TSHR* story, we decided to exclude it from our manuscript. We have deleted the corresponding sentences in the figure legend on page 36.

Reviewer #2 (Remarks to the Author):

Dear editor,

Even though I'm used to review many articles per year, I must say that I'm not a specialist of the specific question asked within the present study. Also, I'm not able to evaluate the entire methods used. Nevertheless, I can give a broad overview of the article from a "naive" perspective; it is therefore very important that the MS must be evaluated by specialists able to assess the entire article.

Overall, I think that the question asked is relevant and interesting. I'm working on fish reproduction for the past 15 years, and for sure timing of reproduction is an important issue. The authors chose a fish species displaying both spring and autumn spawning, the Atlantic herring. Indeed, most temperate fish species spawn once a year (not the case for tropical fish species).

1, I was wondering whether the authors look at rainbow trout (*Oncorhynchus mykiss*), because these species also spawn at different period of time (some domesticated and selected strains); I'm unsure whether natural populations are able to do so, but I would say yes. I do not know whether similar works were performed on that species, but could be interesting to look at either for the present work or further research?

Response: Thank you for the very insightful comment. We haven't looked at the spawning season in details in the trout. It seems that rainbow trout spawn in the spring while brown trout are autumn spawners, it's definitely worthwhile to explore the molecular basis resulting in this difference in the spawning time in these species. As far as we know, our study is the first to investigate the underlying mechanism of seasonal reproduction within a species, which could be a good example for similar studies in other fishes with different spawning seasons within species. In fact, we are in the process to study another salmonid which has both spring- and autumn-spawning populations. We will start the assembly of that genome the coming weeks and expect to have population data later this year. It would be very interesting to compare the results with those presented for the herring in this paper.

As regards whether natural populations are able to spawn at different seasons in nature, we don't have an answer for it now, it would be interesting to revisit the literatures and records for the spawning time for different populations or subspecies within a species. However, we do believe the existence of plasticity in the spawning time in seasonal breeders. For example, it's a common practice to apply photoperiod regimes in aquaculture to regulate fish reproduction, such as Atlantic salmon. In addition, exposing the long day breeder Japanese medaka to summer condition (long photoperiod

and warm temperature) in the laboratory can induce the spawning even in the winter season.

2, Line 50; there are several articles for fish that clearly stated/demonstrated this, you may add Wootton, R. J. (1999). Ecology of Teleost Fishes, 2nd edn. Dordrecht: Kluwer Academic Publishers. ;Teletchea F, Fontaine P (2010) Comparison of early life-stage strategies in temperate freshwater fish species: trade-offs are directed towards first feeding of larvae in spring and early summer. Journal of Fish Biology (2010) 77, 257–278.

Response: Thank you for the suggestion, we have added these two references in our revised manuscript on line 51.

3, In the results, how do you determine “a clear loss of heterozygozy” on the fig 1 (line 101). It is only visually. For me it is indeed quite “obvious” for spring spawning but much less for autumn spawners?

Response: As explained by the comments to reviewer 1 we have replaced this analysis based on SNP data with individual whole genome sequence data, and added a statistical test documenting the highly significant drop in heterozygosity at the *TSHR* locus.

4, The results are very clear and dense (the quantity and quality of data produced is impressive). The discussion is also very clear. I suggest, but you may not take into account this remark (of for further studies), to look at the aquaculture field because what you are looking at could be very interesting for this community as well, notably how the spawning season is “genetically determinate” and how it could be modified (and therefore how the genomic regions might be impacted?).

Response: We completely agree that the results presented here may have important implications in aquaculture and we have modified the “Concluding remarks” and added this sentence:

“The results may also have important implications in aquaculture since gene editing may be used to manipulate the spawning season.”

The materials and methods are clear; yet I’m not enough competent to evaluate in detail what it is indicated. Overall, I consider that the MS is suitable for publication.

Response: Thank you for the positive comments.

Minor comments

1, L70: please take a look at this sentence, might need to be rewritten?

Response: Thank you for the suggestion, we have rewritten the sentence on line 75.

2, L156: please add the scientific name of Pacific herring

Response: Thank you for pointing it out, we have added the scientific name of Pacific herring on line 181.

3, L199: this suggests

Response: Thank you for pointing it out, we have made the change on line 213 in the revised manuscript.

4, L865: this is unclear to what this belongs “d) Relative expression of herring opsin genes in BR and BSH. Y axis 866 shows gene expression values as log₂-transformed TMM-normalized counts-per-million 867 (CPM). A total of 6-7 samples were included in each tissue group (N = 14).” There are no “d” in that Figure ?

Response: Since we think these opsin expression data is not much related to our *TSHR* story, we decided to exclude it from our manuscript. We have deleted the corresponding sentences in the figure legend on page 36.

5, L864: I propose to replace 1 by 100 because the Y-axis is from 0 to 150. It is unclear how I can read this figure, for instance F4A, this imply that the expression level in BSH is 20 fold the expression in gonad? Sorry I'm not very used to this kind of presentation. Might be a stupid remark.

Response: Thank you for this comment. In our tissue profiling experiment, Ct value was first normalized to the housekeeping gene *ACTIN*, then the average expression for each gene in the gonad was assumed to be 1 for the subsequent calculation of the relative expression in other tissues. We have added this information in the method section "Tissue expression profiles by quantitative PCR" on pages 21. The reviewer's interpretation is correct that *TSHR* expression in BSH is about 20-fold higher than the expression in gonad as shown in Figure 4A. Since it's a relative expression analysis, we don't think that replacing 1 by 100 would make big difference in the interpretation of result. Therefore, we respectfully disagree with this comment.

6, L231: you can add the scientific name of sea bass

Response: Thank you for the suggestion, we have added the scientific name of sea bass on line 294.

We look forward to hearing from you regarding our submission and to respond to any further questions and comments you may have.

Sincerely,

Leif Andersson
Junfeng Chen

REVIEWERS' COMMENTS:

Reviewer #1 (Remarks to the Author):

First, I want to apologize to the editor and authors for the delay in returning this review. I found that the authors have addressed satisfactorily most of the comments I had on the first version of this manuscript. It reads very well now and all the pieces of information are well connected. I have two remaining comments that I feel need to be addressed.

First, although the authors are careful in most of the text, the title and some sentences in the abstract imply that the study has identified a direct link between variation in spawning season and allelic variants. Although I agree that they have strong evidence pointing in this direction, I think language has to be toned down. For example, the title states: "Functional differences between TSHR alleles **mediate** variation in spawning season". Further, line 43 states "...non-coding variants at the TSHR locus **regulate** seasonal reproduction in herring". This study shows that the allelic variants in at least one of polymorphic sites of the TSHR locus have functional consequences by affecting its constitutive cAMP signaling. This is very promising, but short of demonstrating that these variants mediate variation in spawning season. As mentioned in other parts of the manuscript, several experiments are still needed to show a causal effect of these variants in timing of reproduction, including evidence for some intermediate mechanisms such as differential TSHR expression, DIO2 expression, variation in circulating T3, and the effect of T3 in timing of reproduction. Indeed, along the text the authors do acknowledge this: e.g., Line 366: "How such an observed difference in basal signaling capacity affects the physiology and development of fishes is not known so far" and line 386 "However, we have not yet been able to compare the temporal profile of TSHR expression for the spring and autumn".

Second, I appreciate the effort to clarify the analyses of selective sweeps. In this version, the authors tested for evidence of selective sweeps in the TSHR locus by comparing heterozygosity in the region of interest versus the background heterozygosity for the different variants. Although this extends my field of expertise, my understanding is that it is better to test for selective sweeps by comparing different statistics for the region of interests than solely relying in heterozygosity. For example, comparing Tajima's D or more complex test like those performed in Sweepfinder (for a recent method published in the journal see <https://www.nature.com/articles/s42003-018-0085-8>, although it has been suggested that this method might report regions of low differentiation as outliers and proper masking is required). Given the results reported by the authors, I don't think much would change, but I was wondering if the authors could attempt one of these methods or explain why they decided on basing their conclusions solely in heterozygosity. Another thing that could be clarified is why they based their analyses in 500 SNPs windows. I guess it depends on the LD blocks, but this is not reported (it is just stated that there is strong linkage disequilibrium). Finally, Figure 1b and c may benefit from including a close up of the TSHR, it would be useful for the reader as it is hard to see much in these chromosome-level figures.

Line 203: BSH meaning should be described here, as it is first mentioned here, rather than in line 214.

Line 213-223: The authors have clarified in their rebuttal letter that the fish used for the expression data are spring spawners. I see why they think it might not be completely needed to report in this paragraph this information, but because different analyses use different fish samples, I would recommend including the phenotype of the fish used in this section.

Line 255: a 1.2 rather than a 2.2-fold increase in constitutive cAMP signaling is still being reported here.

Best wishes!

Reviewer #2 (Remarks to the Author):

Dear editor

The authors have taken into account all my minor comments. I carefully read the MS and was not able to find any minor mistakes left. Again, even though I'm not able to understand all molecular analyses performed, I enjoyed reading the MS and think it is now suitable for publication.

Response to the comments from Reviewers

We thank the Reviewers again for your time and valuable comments on our manuscript. Here is a point-by-point response, in blue, to the reviewers' comments and concerns. All the line numbers refer to the revised manuscript file.

Reviewer #1 (Remarks to the Author):

First, I want to apologize to the editor and authors for the delay in returning this review.

I found that the authors have addressed satisfactorily most of the comments I had on the first version of this manuscript. It reads very well now and all the pieces of information are well connected.

Response: Thank you for the positive comments.

I have two remaining comments that I feel need to be addressed.

First, although the authors are careful in most of the text, the title and some sentences in the abstract imply that the study has identified a direct link between variation in spawning season and allelic variants. Although I agree that they have strong evidence pointing in this direction, I think language has to be toned down. For example, the title states: "Functional differences between TSHR alleles mediate variation in spawning season". Further, line 43 states "...non-coding variants at the TSHR locus regulate seasonal reproduction in herring". This study shows that the allelic variants in at least one of polymorphic sites of the TSHR locus have functional consequences by affecting its constitutive cAMP signaling. This is very promising, but short of demonstrating that these variants mediate variation in spawning season. As mentioned in other parts of the manuscript, several experiments are still needed to show a causal effect of these variants in timing of reproduction, including evidence for some intermediate mechanisms such as differential TSHR expression, DIO2 expression, variation in circulating T3, and the effect of T3 in timing of reproduction. Indeed, along the text the authors do acknowledge this: e.g., Line 366: "How such an observed difference in basal signaling capacity affects the physiology and development of fishes is not known so far" and line 386 "However, we have not yet been able to compare the temporal profile of TSHR expression for the spring and autumn".

Response: We have modified the title and replaced "mediate" with "associate with" and modified the text in the abstract to acknowledge that the causal relationships are not yet fully understood.

Second, I appreciate the effort to clarify the analyses of selective sweeps. In this version, the authors tested for evidence of selective sweeps in the TSHR locus by comparing heterozygosity in the region of interest versus the background heterozygosity for the different variants. Although this extends my field of expertise, my understanding is that it is better to test for selective sweeps by comparing different statistics for the region of interests than solely relying in heterozygosity. For example, comparing Tajima's D or more complex test like those performed in Sweepfinder (for a recent method published in the journal see <https://www.nature.com/articles/s42003-018-0085-8>, although it has been suggested that this method might report regions of low differentiation as outliers and proper masking is required). Given the results reported by the authors, I don't think much would change, but I was wondering if the authors could attempt one of these methods or explain why they

decided on basing their conclusions solely in heterozygosity. Another thing that could be clarified is why they based their analyses in 500 SNPs windows. I guess it depends on the LD blocks, but this is not reported (it is just stated that there is strong linkage disequilibrium). Finally, Figure 1b and c may benefit from including a close up of the TSHR, it would be useful for the reader as it is hard to see much in these chromosome-level figures.

Response: We have modified this analysis and assessed nucleotide diversities, Tajima's D and delta allele frequencies. We have also added a close up of the TSHR region as suggested by the reviewer. However, the detection of a sweep signal is not critical for this study, we included these data to introduce the readers to the underlying population data. A sweep signal is present for more recent selection events but such signals disappear over time as recombination breaks up linkage between the target of selection and flanking regions. The most important aspect here is the highly significant genetic differentiation between spring- and autumn-spawners (Figure 1a) and that this differentiation peaks exactly at the TSHR locus (Figure 1b). This is better communicated with the revised analysis.

Line 203: BSH meaning should be described here, as it is first mentioned here, rather than in line 214.

Response: Thank you for pointing this out, we have defined the BSH on line 234 and deleted the definition on line 246 in the revised manuscript.

Line 213-223: The authors have clarified in their rebuttal letter that the fish used for the expression data are spring spawners. I see why they think it might not be completely needed to report in this paragraph this information, but because different analyses use different fish samples, I would recommend including the phenotype of the fish used in this section.

Response: Thank you for the suggestion, we have added the fish information on line 254 in the revised manuscript.

Line 255: a 1.2 rather than a 2.2-fold increase in constitutive cAMP signaling is still being reported here.

Response: Thank you for pointing it out, we have changed "1.2-fold" to "2.2-fold" on line 292 in the revised manuscript.

Reviewer #2 (Remarks to the Author):

Dear editor

The authors have taken into account all my minor comments. I carefully read the MS and was not able to find any minor mistakes left. Again, even though I'm not able to understand all molecular analyses performed, I enjoyed reading the MS and think it is now suitable for publication.

Response: Thank you for the positive feedback on this version of our manuscript.

We look forward to hearing from you regarding this revised version and to respond to any further questions you may have.

Sincerely,

Leif Andersson

Junfeng Chen